# Remote control of microtubule plus-end dynamics and function from the minus-end

Xiuzhen Chen[1†], Lukas A Widmer[2,3†], Marcel M Stangier[4], Michel O Steinmetz[4,5], Jörg Stelling[2*], Yves Barral[1*]

[1]Institute of Biochemistry, ETH Zürich, Zurich, Switzerland; [2]Department of Biosystems Science and Engineering, ETH Zürich, SIB Swiss Institute of Bioinformatics, Basel, Switzerland; [3]Systems Biology PhD Program, Life Science Zurich Graduate School, Zurich, Switzerland; [4]Laboratory of Biomolecular Research, Department of Biology and Chemistry, Paul Scherrer Institut, Villigen, Switzerland; [5]Biozentrum, University of Basel, Basel, Switzerland

**Abstract** In eukaryotes, the organization and function of the microtubule cytoskeleton depend on the allocation of different roles to individual microtubules. For example, many asymmetrically dividing cells differentially specify microtubule behavior at old and new centrosomes. Here we show that yeast spindle pole bodies (SPBs, yeast centrosomes) differentially control the plus-end dynamics and cargoes of their astral microtubules, remotely from the minus-end. The old SPB recruits the kinesin motor protein Kip2, which then translocates to the plus-end of the emanating microtubules, promotes their extension and delivers dynein into the bud. Kip2 recruitment at the SPB depends on Bub2 and Bfa1, and phosphorylation of cytoplasmic Kip2 prevents random lattice binding. Releasing Kip2 of its control by SPBs equalizes its distribution, the length of microtubules and dynein distribution between the mother cell and its bud. These observations reveal that microtubule organizing centers use minus to plus-end directed remote control to individualize microtubule function.
DOI: https://doi.org/10.7554/eLife.48627.001

*For correspondence:
joerg.stelling@bsse.ethz.ch (JS);
yves.barral@bc.biol.ethz.ch (YB)

[†]These authors contributed equally to this work

Competing interests: The authors declare that no competing interests exist.

## Introduction

Microtubules are hollow cylinders formed by the polymerization of αβ-tubulin dimers. Throughout eukaryotes, they drive mitotic chromosome segregation, form cilia and flagella, and function in cell polarity (*Desai and Mitchison, 1997*). They also provide tracks to properly position organelles and to transport diverse cargos through the cell (*Hirokawa et al., 2009*; *Reck-Peterson et al., 2018*). Polarity that results from the head-to-tail polymerization of tubulin dimers is a major prerequisite for microtubule function. Microtubules expose α-tubulin at one end, called the minus-end and β-tubulin at the opposed end, the plus-end (*Mitchison, 1993*). The minus-end of most microtubules is stabilized by a γ-tubulin cap, which tethers microtubule minus-ends to microtubule organizing centers (MTOCs) such as centrosomes in animal cells and spindle pole bodies (SPBs) in fungi and many protists (*Sanchez and Feldman, 2017*). The microtubule plus-end grows away from the MTOC and explores the cellular space by dynamically alternating between periods of growth and shrinkage, a process called dynamic instability (*Mitchison and Kirschner, 1984*). This search function allows microtubules to reach attachment sites, for example, kinetochores on chromosomes or various cortical sites (*Akhmanova and Steinmetz, 2008*; *Kirschner and Mitchison, 1986*).

Remarkably, although all microtubules of a cell are formed from the same pool of tubulin dimers, they can show distinct behaviors and functions. For example, in most cell types at least three kinds

of microtubules contribute to the assembly of the mitotic spindle. According to their distinct roles, these microtubules show different lengths and dynamics (*Wittmann et al., 2001*). For instance, concomitant to kinetochore microtubules shrinking and pulling chromosomes towards the spindle pole, the interpolar microtubules grow, push the spindle poles away from each other and drive spindle elongation. During that time, the astral microtubules that help move the spindle in the cell and position the future cleavage plane, may show different behavior depending on which of the two spindle poles they emanate from (*Shaw et al., 1997*; *Januschke and Gonzalez, 2010*; *Lengefeld et al., 2018*).

To distinctively control the behavior and function of individual microtubules even within a single cellular environment, cells have evolved several mechanisms. A diverse set of microtubule-associated proteins (MAPs), plus-end tracking proteins (+TIPs), and microtubule-dependent motor proteins can differentially associate with microtubules and contribute to their distinctive behaviors and activities (*Akhmanova and Steinmetz, 2008*). In many instances the behavior of a microtubule depends largely on its subcellular environment and the cellular structures that it contacts with its plus-end (capture model; *Kirschner and Mitchison, 1986*). Furthermore, in many cell types post-translational modifications of tubulin along microtubules modulate both the microtubules' stability and their affinity for the above regulatory factors (tubulin code; *Gadadhar et al., 2017*). However, in other cells, such as mitotic cells, the behavior of microtubules seems to also depend on the MTOC they emanate from (*Januschke and Gonzalez, 2010*; *Lengefeld et al., 2018*; *Yamashita et al., 2007*; *Gasic et al., 2015*; and below). For example, in many asymmetrically dividing cells one of the two asters forms microtubules that are less dynamic than the other one. However, how MTOCs specify the plus-end behavior of the microtubules emanating from them remains unknown.

The differential control of microtubule dynamics and function observed in mitotic yeast cells offers an ideal model system for addressing this question. The yeast mitotic spindle assembles within the nucleus between two SPBs embedded in the nuclear envelope. The SPBs also nucleate cytoplasmic microtubules that facilitate the movement of the spindle to the mother-bud neck and its alignment with the mother-bud axis. Thereby they ensure that the mother cell and the bud faithfully inherit each one copy of the genome upon division. Remarkably, throughout yeast preanaphase the two asters of cytoplasmic microtubules are morphologically and functionally distinct (*Shaw et al., 1997*; *Lengefeld et al., 2018*). The SPB oriented towards the mother cell (m-SPB), which is the newly synthesized SPB (young SPB) in virtually all cells, nucleates few and short microtubules (m-microtubules). In contrast, the bud-oriented SPB (b-SPB), which is inherited from the previous mitosis (old SPB), forms long microtubules that extend into the bud (b-microtubules) (*Shaw et al., 1997*; *Lengefeld et al., 2018*). The b-microtubules, specifically, transport the minus-end directed motor protein dynein to their plus-ends and deliver it to the bud cortex (*Moore et al., 2009*). Activated at anaphase onset, dynein at the bud cortex pulls on b-microtubules and facilitates the entry of the b-SPB and the associated daughter nucleus through the bud neck, into the bud (*Adames and Cooper, 2000*; *Tang et al., 2012*). How yeast cells assign their different functions to m- and b-microtubules, although these microtubules are polymerized from the same pool of soluble tubulin dimers, is unclear. Moreover, there is no evidence that yeast modifies tubulin along microtubules (*Drummond et al., 2011*; *Uchimura et al., 2006*), indicating that microtubule differentiation inside the cell depends exclusively on the differential recruitment of regulatory proteins. We know, however, little about which are the relevant regulators and how their recruitment is controlled.

Recent work has established that many kinesins, a particularly rich family of microtubule-dependent motor proteins, regulate microtubule dynamics (*Howard and Hyman, 2007*; *Akhmanova and Steinmetz, 2015*) in addition to their function in cargo transport. Paradigmatic examples include the yeast kinesins Kip2 and Kip3. While Kip3 promotes microtubule catastrophe (*Varga et al., 2006*; *Arellano-Santoyo et al., 2017*; *Varga et al., 2009*; *Gupta et al., 2006*), that is the switch from plus-end growth to shrinkage, Kip2, which transports dynein to the plus-end of microtubules (*Moore et al., 2009*), inhibits catastrophe and promotes microtubule elongation (*Hibbel et al., 2015*; *Cottingham and Hoyt, 1997*; *Huyett et al., 1998*; *Carvalho et al., 2004*). Thus, these kinesins are excellent candidates for controlling the behavior of individual microtubules. In order to determine whether they contribute to specifying the distinct behavior of m- and b-microtubules in yeast, we characterized their localization in preanaphase cells and dissected the mechanisms controlling their distribution.

## Results

### Kip2 distribution relies on different mechanisms in vivo and in vitro

In order to characterize the distribution of Kip2 and Kip3 along cytoplasmic microtubules in yeast cells, three copies of the super-folder green fluorescent protein (3xsfGFP) were fused to the C-terminus of each endogenous motor protein. The SPBs were visualized by fusing mCherry to the SPB component Spc42. All fusion proteins were functional, as determined by spot growth assays with cells expressing these fusion genes in mutant backgrounds that do not support growth in the absence of either Kip2 or Kip3 function (*Figure 1—figure supplement 1a*). To quantify the distributions of motile kinesins, we used a spinning disk microscope to achieve high acquisition speed and recorded images at 17 Z-sections separated by 0.24 µm increments in 1.07 s and 1.78 s for the GFP and mCherry channels, respectively. We performed line scanning analysis of cytoplasmic microtubules, aligned and averaged the signal obtained for each reporter for large collections of images of preanaphase cells (n ≥ 279 per strain), and computed the distribution of each kinesin along cytoplasmic microtubules as a function of microtubule length.

In agreement with published in vitro data (*Varga et al., 2006*; *Varga et al., 2009*), Kip3-3xsfGFP intensity increased along cytoplasmic microtubules, the protein being barely detectable at their minus-ends and reaching maximal levels at the plus-ends (*Figure 1a*; note that the abundant GFP fluorescence around the SPB originated from Kip3-3xsfGFP on the nuclear spindle microtubules). Furthermore, Kip3 amounts at microtubule plus-ends increased linearly with microtubule length (*Figure 1a* and *Figure 1—figure supplement 1b*). Since Kip3 triggers plus-end catastrophes in a manner that depends on the local concentration at the plus-end (*Varga et al., 2006*; *Varga et al., 2009*), our analysis indicates that Kip3 is a length-dependent microtubule depolymerase in vivo, consistent with the antenna model established in vitro (*Varga et al., 2006*; *Varga et al., 2009*; *Gupta et al., 2006*).

Strikingly, although in vitro the patterns of Kip2 and Kip3 distribution along microtubules are very similar (*Hibbel et al., 2015*), the distribution of Kip2 followed a different pattern in vivo: the signal for Kip2-3xsfGFP failed to increase along microtubules. Instead, it remained flat and above-zero from the minus-end of the microtubule up to the vicinity of its plus-end, where it peaked locally (*Figure 1b* and *Figure 1—figure supplement 1c*). This distinct distribution of Kip2-3xsfGFP in vivo was confirmed by further technical replicates of the line scan analysis (*Figure 1—figure supplement 1d*). Neither the level of the protein along the microtubule nor the height of its plus-end peak varied with microtubule length. The sharp discrepancy between the distributions of Kip2 in vivo and in vitro suggests that a mechanism distinct from the one suggested in vitro (*Hibbel et al., 2015*) controls Kip2 distribution in living yeast cells.

### Modelling suggests recruitment of Kip2 to the microtubule minus-end

In vitro, the distribution of Kip2 and Kip3 relies on their ability (1) to run towards the microtubule plus-end faster than the microtubule grows, (2) to run over long distances without falling off (processivity), and (3) to land and initiate their runs at any site on the microtubule lattice. Since long microtubules have more lattice sites than short ones, they collect more motor molecules (*Varga et al., 2006*; *Hibbel et al., 2015*). To understand how Kip2 reaches its unique distribution in vivo, we modeled the distribution profile of an idealized plus-end-directed motor on static microtubule filaments (see Appendix 1 for details) as a function of four motor parameters that determine on-rate ($k_{on}$, rate constant of landing and starting a run at each location on the microtubule lattice), off-rate ($k_{off}$, rate of falling off from the microtubule lattice), stepping speed (with rate $k_{step}$), and rate of detachment from microtubule plus-ends ($k_{out}$) (*Figure 2a*). This model easily explains the accumulation of Kip2 at microtubule plus-ends by a low plus-end detachment rate. Analytically solving this model predicts a flat distribution only in special cases, such as when the on-rate is zero (no motor on the microtubule lattice) or the motor speed is zero (inactive motors binding everywhere; see Appendix 1 for details). However, none of these cases was compatible with our experimental observations. For example, Kip2 moves processively and at least as fast as Kip3, both in vitro (*Varga et al., 2006*; *Varga et al., 2009*; *Hibbel et al., 2015*; *Roberts et al., 2014*) and in vivo (see below and *Figure 1—figure supplement 1ab*).

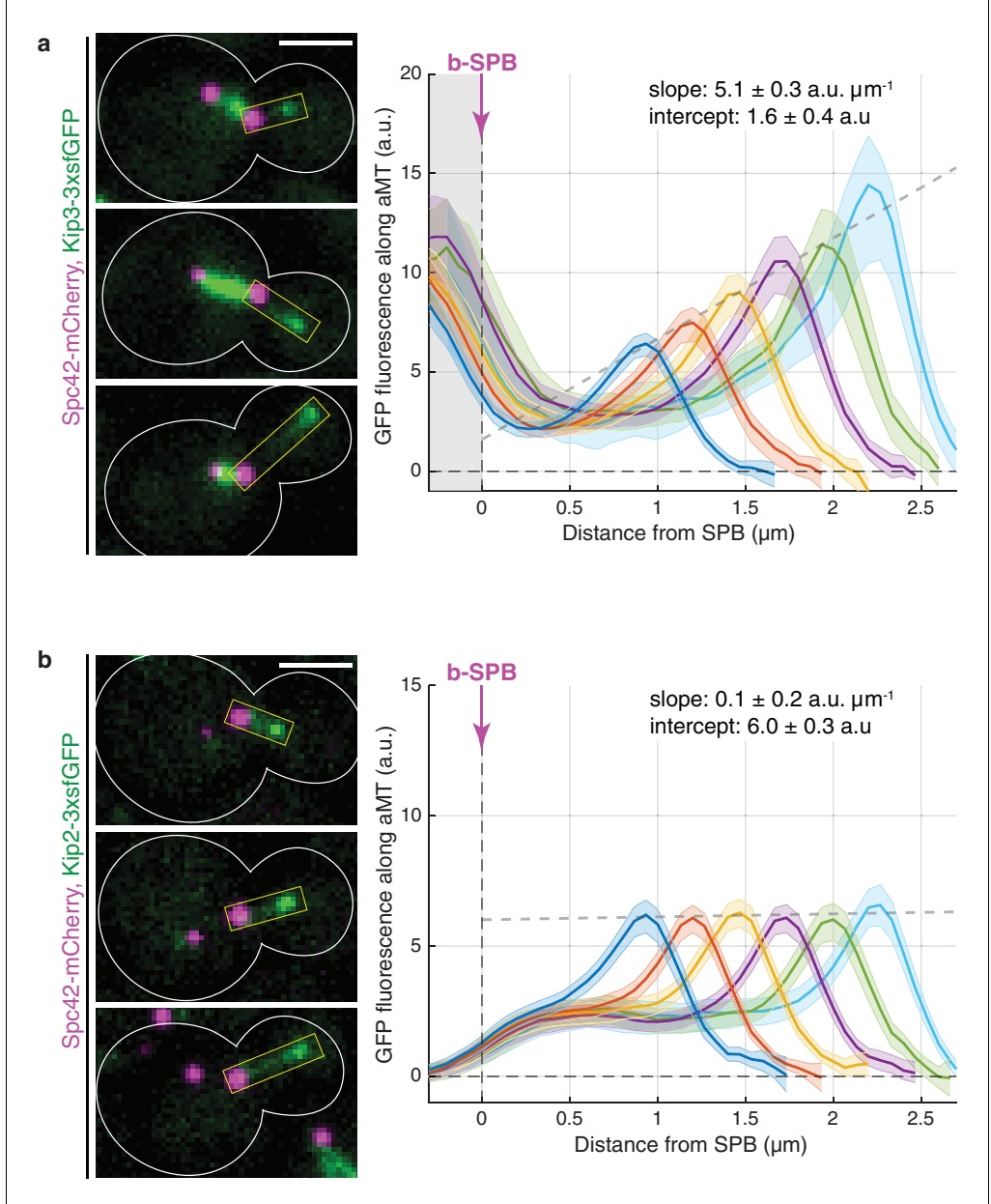

**Figure 1.** Kinesins Kip3 and Kip2 exhibit distinct localization patterns along microtubules in vivo. (a, b) Representative images (left) and quantifications (right) of fluorescence intensities (a.u.) from endogenous Kip3-3xsfGFP (a) and Kip2-3xsfGFP (b) along preanaphase astral microtubules (aMTs; boxed areas). Signals were aligned to b-SPBs using the peak of Spc42-mCherry (magenta) intensity and binned by microtubule length (2-pixel = 266.7 nm bin size). Colored lines show mean Kip2/3-3xsfGFP fluorescence per bin and shaded areas represent 95% confidence intervals for the mean. Gray dashed lines denote weighted linear regressions for the mean GFP fluorescence on plus-ends over all bins. The area on the left of the vertical dashed line passing through x = 0 in (a) marks Kip3-3xsfGFP fluorescence inside nuclei. Scale bars, 2 μm. 10 ≤ n ≤ 130 per bin. See also *Figure 1—figure supplement 1*.

DOI: https://doi.org/10.7554/eLife.48627.002

The following figure supplement is available for figure 1:

**Figure supplement 1.** Functional analysis of endogenously tagged proteins and reproducibility of the kinesin distribution analysis in vivo.

DOI: https://doi.org/10.7554/eLife.48627.003

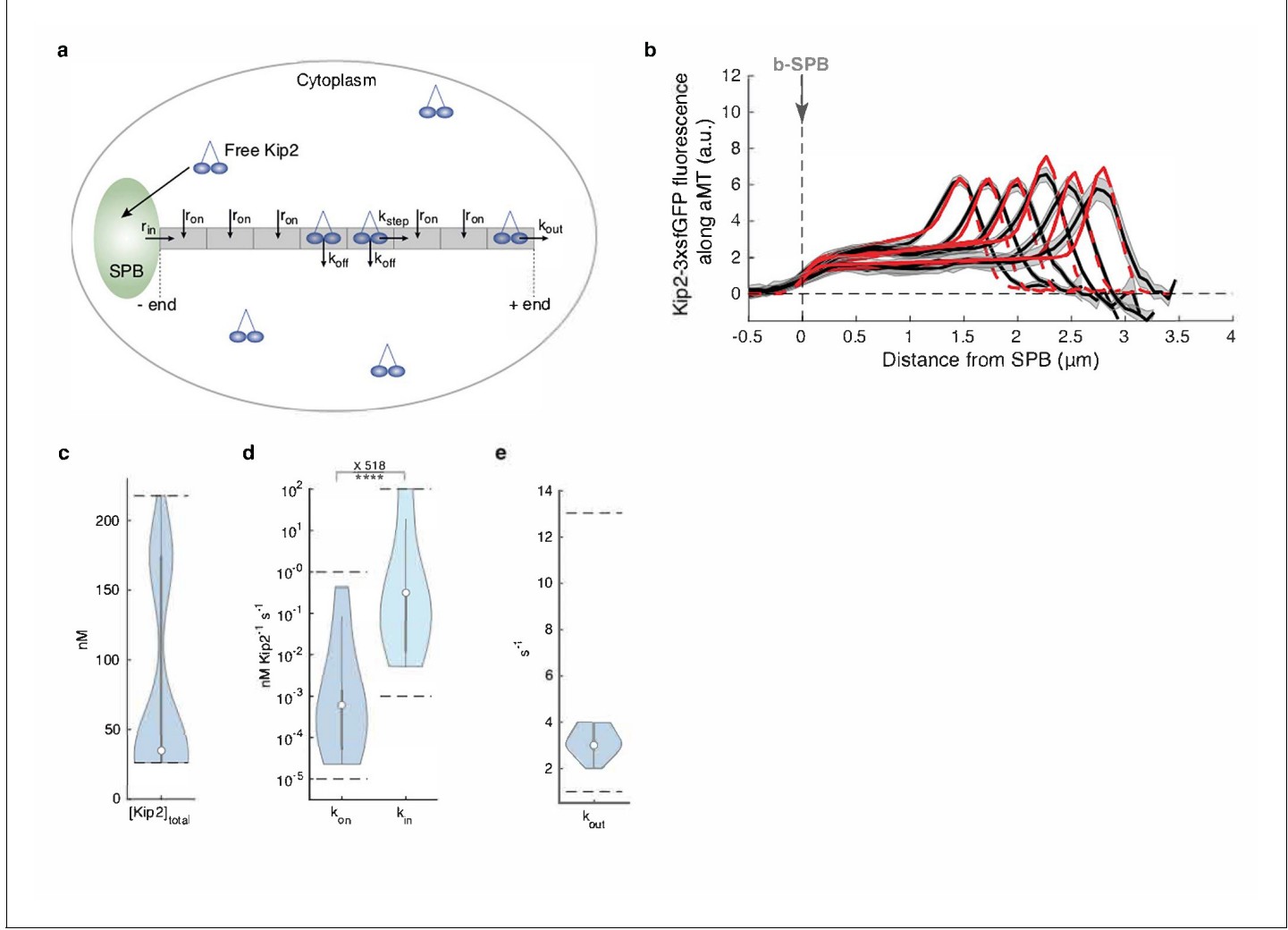

**Figure 2.** Mathematical model predicts recruitment of Kip2 to the microtubule minus-end. (**a**) Schematic of the mathematical model. Free Kip2 (with concentration $[\text{Kip2}]_{\text{free}} \leq [\text{Kip2}]_{\text{total}}$) binds to the microtubule minus-end anchored at the SPB with rate $r_{\text{in}} = k_{\text{in}}[\text{Kip2}]_{\text{free}}$ if the minus-end site is free, and to any free lattice site with rate $r_{\text{on}} = k_{\text{on}}[\text{Kip2}]_{\text{free}}$. A bound motor can detach with rate $k_{\text{off}}$, and it can advance with rate $k_{\text{step}}$ if the next site towards the plus-end is free. At the plus end, the motor detaches with a different rate, $k_{\text{out}}$. (**b**) Experimental Kip2-3xsfGFP fluorescence (a.u.) mean profile (black) and standard error (gray) with respective mean in silico model fits (red) for microtubules binned by length. Red dashed lines past plus-end and SPB indicate model extrapolations without support by data. (**c**) Likelihood of total Kip2 concentration $[\text{Kip2}]_{\text{total}}$ estimated from fit in (**b**). (**d**) Likelihood of on rate constant $k_{\text{on}}$ and in rate constant $k_{\text{in}}$ estimated from in silico model fit in (**b**). Statistical significance for $k_{\text{in}} > k_{\text{on}}$, ****, (p<5·10$^{-5}$) determined by sampling from the likelihood (see Appendix 1), difference in median as indicated. (**e**) Likelihood of out rate $k_{\text{out}}$ constant estimated from in silico model fit in (**b**). For in silico model parameter estimate plots (cde), median parameter values are indicated as circles, inter-quartile range (*IQR*) by thick gray bars and $1.5 \times IQR$ by thin gray bars. Kernel density estimates are computed from 20'000 samples from the likelihood function. Sampled parameter ranges are indicated by dashed black lines. See also *Figure 2—figure supplement 2*, *Figure 2—figure supplement 1*, and *Figure 2—figure supplement 3*.

DOI: https://doi.org/10.7554/eLife.48627.004

The following figure supplements are available for figure 2:

**Figure supplement 1.** Median parameter value model simulations for static and growing microtubules.

DOI: https://doi.org/10.7554/eLife.48627.005

**Figure supplement 2.** Measurements of kinesin movement speeds, as well as the speeds of b-microtubule growth and shrinkage in living cells.

DOI: https://doi.org/10.7554/eLife.48627.006

**Figure supplement 3.** Concentration and out rate estimates for *bfa1Δbub2Δ* and Kip2-S63A strains.

DOI: https://doi.org/10.7554/eLife.48627.007

We reasoned that the flat profile of Kip2 on microtubule shafts resembled the average distribution of vehicles on a track between entry and exit sites at a constant speed and entry rate. Therefore, we included a minus-end entry site in our model by adding a minus-end loading rate parameter (rate constant $k_{in}$, *Figure 2a*). Analytically solving for a flat distribution shows that the only biologically feasible solution for this model is obtained when all Kip2 is recruited to the microtubule at its minus-end, and Kip2 unbinding on the lattice ($k_{off}$) is zero (see Appendix 1). Next, we numerically estimated the parameters for this model using the experimental concentration profiles of Kip2 on microtubules in vivo. Using these estimates, the model with static microtubules faithfully represented the flat motor distributions along microtubules of all length categories (*Figure 2b*), and we obtained the same qualitative predictions for a model with growing microtubules as a control (see Appendix 1 and *Figure 2—figure supplement 1*). Importantly, the estimated Kip2 concentration and the on-rate constant were close to published values (*Hibbel et al., 2015*; *Roberts et al., 2014*) (*Figure 2cd*, *Figure 2—figure supplement 1a*, see Appendix 1 for details), lending support to the model. The estimated motor parameters predicted minus-end recruitment (in-rate) and plus-end detachment (out-rate) of the same magnitude, and low Kip2 recruitment on the lattice (low on-rate; *Figure 2de*). Importantly, the median in-rate constant was approximately 500 times higher than the median on-rate constant at each lattice site ($p < 5 \cdot 10^{-5}$, see Appendix 1 for details). Therefore, both the analytical and the numerical modeling results point towards Kip2 initiating its runs on the microtubule predominantly from its minus-end, that is, at or near the SPB from which the microtubule emanates, as an explanation for the unique distribution of Kip2 along microtubules in vivo.

## Kip2 runs start from SPBs

To test this possibility, we next analyzed where Kip2 initiates its runs. First, we recorded the movement of Kip2-3xsfGFP along microtubules in vivo with high temporal resolution time-lapse series, using Spc72-GFP as an SPB marker. As previously reported (*Carvalho et al., 2004*), Kip2-3xsfGFP appeared as moving speckles along cytoplasmic microtubules (*Video 1*). For quantitative analysis, we collected a sufficient number (n = 45) of the rare preanaphase cells in which the cytoplasmic microtubules stayed relatively still in the imaging plane throughout the imaging time (85.7 s, 80 frames); this allowed us to generate kymographs that cover the complete time window (*Figure 3a* and *Figure 2—figure supplement 2a*). Overall, these kymographs clearly established that the Kip2 signal was indeed evenly distributed along microtubules, compared to Kip3-3xsfGFP kymographs, where the kinesin signal became visible only progressively on the microtubules, towards the plus-end. Despite the strong noise associated with these kymographs, speckles were identifiably for both Kip2- and Kip3-3xsfGFP and accounted for most of the signal. Kip2 speckle movement speed in these kymographs established that, in vivo, Kip2 runs along microtubules at an average speed of $6.3 \pm 2.1$ µm min$^{-1}$ (mean ± S.D., n = 192 speckles, *Figure 2—figure supplement 2b*), which is more than four times faster than the mean microtubule growth speed under the same conditions ($1.4 \pm 1.1$ µm min$^{-1}$, mean ± S.D., n = 250 growth phases, *Figure 1—figure supplement 1cd*). Furthermore, in all our experimental Kip2-3xsfGFP kymographs (n = 45), all unambiguous trains of motors (yellow arrowheads, *Figure 3a*; n = 192) moving along the imaged microtubules started from SPBs and ended at the microtubule plus-ends, as predicted by the model (*Figure 3a*, *Video 1*). Tracks that appeared to stop in the middle of the microtubule (red arrowhead, *Figure 3a*) resulted from microtubules moving out of the focus when looking back at the source movie. As another mean of visualizing the movement of single Kip2-3xsfGFP speckles, we performed line scans of microtubules from time-lapse series. As shown in *Figure 3b*, all the fluorescence speckles that we analyzed this way (n = 9) also departed from the SPB, traveled along the microtubule shaft and arrived at the plus-end. In contrast, under the same imaging conditions, Kip3-3xsfGFP rarely appeared as

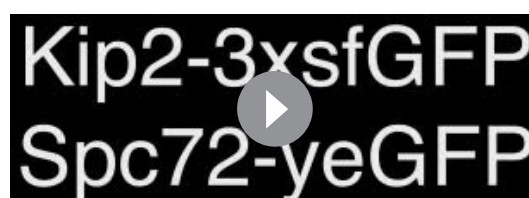

**Video 1.** A representative time-lapse movie shows Kip2-3xsfGFP molecules appearing as speckles from the SPB and moving along the cytoplasmic microtubule towards its plus-end. All Kip2-3xsfGFP speckles reach the plus-end. The movie consists of 80 frames that were taken every 1.07 s and the frame speed is sped up by 3-fold for better visualization. Scale bar, 2 µm.
DOI: https://doi.org/10.7554/eLife.48627.011

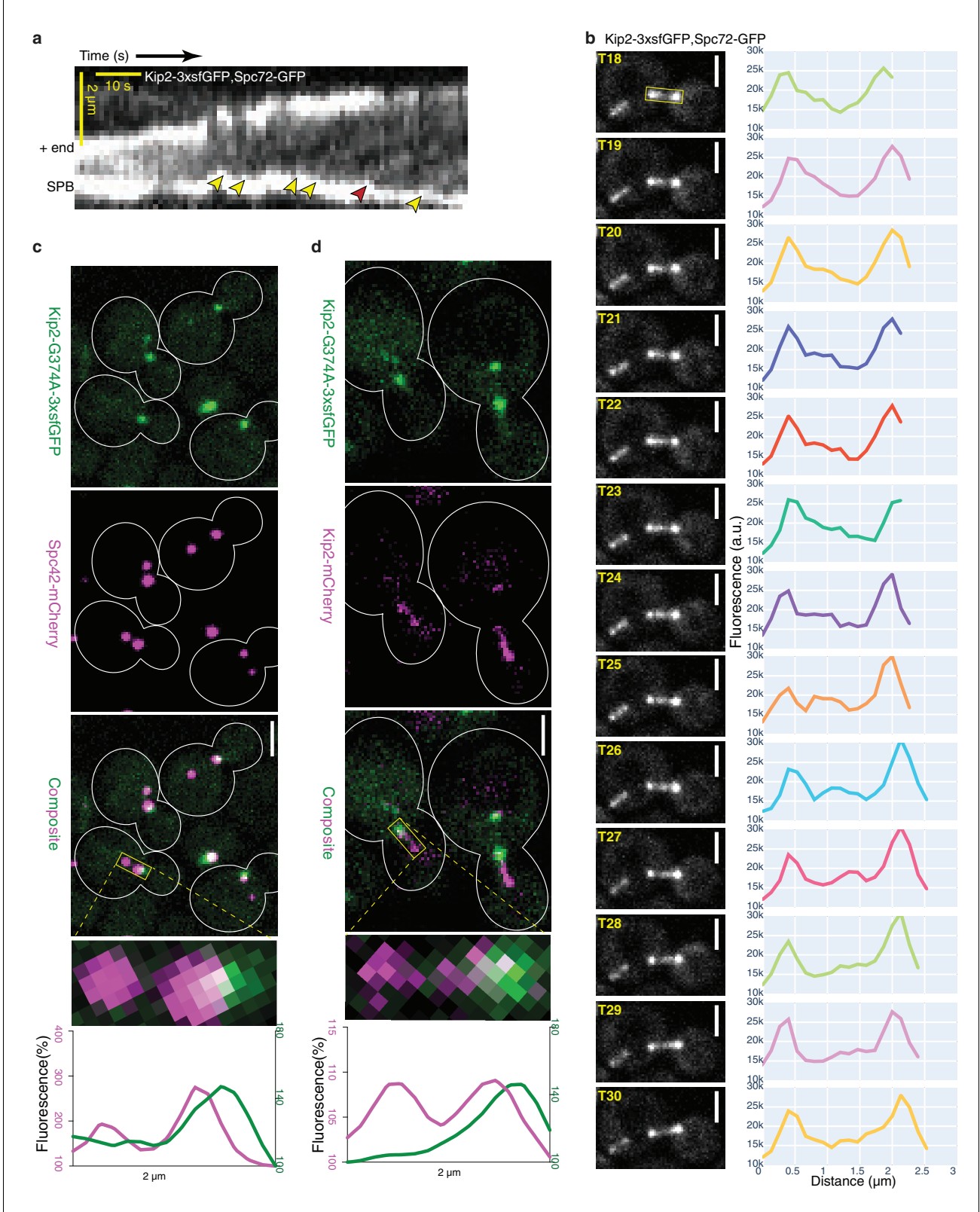

**Figure 3.** Kip2 initiates its runs from SPBs or their vicinity. (**a**) Representative experimental kymograph showing Kip2-3xsfGFP speckles departing (yellow arrows) from the SPB, visualized with Spc72-GFP. The red arrow marks a Kip2-3xsfGFP speckle that moves towards the part of the microtubule which moves out of focus. (**b**) Images of *Video 1* from time frame 18 to 30 and line scan analysis of the b-microtubule (boxed area) showing a Kip2-3xsfGFP speckle that departs from the SPB (T18–T19), moves along the microtubule shaft (T20–T29), and arrives at the plus-end (T30). (**c**) Representative images

*Figure 3 continued on next page*

*Figure 3 continued*

of preanaphase cells expressing the endogenous ATPase deficient protein Kip2-G374A-3xsfGFP (green) and Spc42-mCherry (magenta). Close-up of a mitotic spindle (boxed area, 2 μm long) and line scan analysis of this area are shown on the right. Scale bars, 2 μm. (d) Representative images of preanaphase heterozygous diploid cells expressing Kip2-G374A-3xsfGFP (green) and the wild type protein Kip2-mCherry (magenta). Close-up (boxed area, 2 μm long) and line scan analysis as in (b). Scale bars, 2 μm. For the line scan analysis in (b,c), GFP (green) and mCherry (magenta) fluorescence intensities were normalized to their background levels, respectively. See also *Video 1*, *Figure 2—figure supplement 2a*, *Figure 3—figure supplement 1* and *Figure 3—figure supplement 2*.

DOI: https://doi.org/10.7554/eLife.48627.008

The following figure supplements are available for figure 3:

**Figure supplement 1.** Sequence alignment of various kinesin motor domains focused on the switch-2 motif (DxxGxE, highlighted in bold) (*Marx et al., 2009*) that is essential for the ATP hydrolysis.

DOI: https://doi.org/10.7554/eLife.48627.009

**Figure supplement 2.** Kip2-G374A-3xsfGFP colocalizes with Spc72-mCherry.

DOI: https://doi.org/10.7554/eLife.48627.010

fluorescent speckles (n = 10, from eight kymographs) and the speckles started moving from random places along microtubules (*Figure 2—figure supplement 2a*, *Video 2*), as expected from Kip3's distribution profile (*Figure 1a*) and in vitro data (*Varga et al., 2006*; *Varga et al., 2009*). These data support the notion that unlike Kip3, the initiation of Kip2 runs is restricted to the minus-ends of microtubules – at least in preanaphase cells.

As a second independent test for whether Kip2 starts its runs from microtubule minus-ends, we aimed at directly visualizing Kip2 loading sites on microtubules. We mutated a conserved glycine, G374 in Kip2 (Kip2-G374A), which is required for ATP hydrolysis in all kinesins (*Figure 3—figure supplement 1*). The ATPase deficient Kip2-G374A-3xsfGFP cannot move along the microtubule following its initial binding and should therefore accumulate at the sites where it is loaded on the microtubules. Strikingly, the Kip2-G374A-3xsfGFP signal accumulated exclusively at one or two point-like structures in preanaphase cells and co-expression of the SPB-marker Spc42-mCherry showed that the mutated motor signal systematically localized to or near SPBs (*Figure 3c*). Closer analysis of the signal's distribution indicated that Kip2-G374A-3xsfGFP was concentrated on the cytoplasmic side of the SPB. Accordingly, Kip2-G374A-3xsfGFP localization extensively overlapped with Spc72-mCherry, the receptor of the γ-tubulin complex on the SPB outer-plaque (*Knop and Schiebel, 1998*) (*Figure 3—figure supplement 2*). This provided further, independent evidence that Kip2 recruitment to microtubules is restricted nearly exclusively to SPBs, that is, at or near the minus-ends of cytoplasmic microtubules.

To confirm that the exclusive localization of Kip2-G374A-3xsfGFP to SPBs was not due to general defects in microtubule organization, owing to Kip2 defects, we also characterized the localization of the protein in the presence of the wild type form of Kip2 by imaging heterozygous diploid cells expressing both the wild type protein fused to mCherry (Kip2-mCherry) and Kip2-G374A-3xsfGFP. In these cells, the Kip2-mCherry protein decorated microtubule shafts, SPBs and plus-ends, as expected. In contrast, the ATPase-deficient, mutant protein remained at or near SPBs (*Figure 3d*). More of Kip2-mCherry localized to SPBs in these strains than in cells that do not express Kip2-G374A, probably due to heterodimerization of the wild type protein with its ATPase deficient mutant. On the opposite, very little if any Kip2-G374A-3xsfGFP was transported to microtubule plus-ends by the wild type protein. We conclude that the heterodimers Kip2/Kip2-G374A are inactive and either stay at their loading site on microtubules, like Kip2-G374A homodimers, or rapidly fall off the microtubule shaft. Moreover, these data established that the landing of Kip2 on microtubules is highly

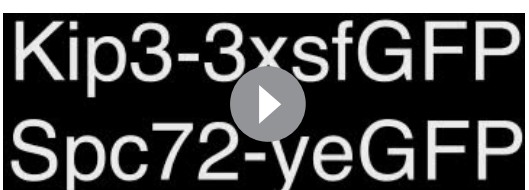

**Video 2.** A representative time-lapse movie shows that Kip3-3xsfGFP molecules are nearly absent from the minus-end of the cytoplasmic microtubule and accumulate along the shaft to reach the maximum intensity on the plus-end. The movie consists of 80 frames that were taken every 1.07 s and the frame speed is sped up by 3-fold for better visualization. Scale bar, 2 μm.

DOI: https://doi.org/10.7554/eLife.48627.012

restricted to SPBs. These data also exclude the possibility of Kip2 being directly recruited to microtubule plus-ends, independent of its motor activity.

## Mother- and bud-directed SPBs recruit different levels of Kip2

Strikingly, quantification of the GFP signal in cells coexpressing Kip2-G374A-3xsfGFP and Spc42-mCherry established that the two SPBs of preanaphase cells did not recruit equal levels of Kip2. b-SPBs localized near the bud neck accumulated approximately four times more of the ATPase-deficient Kip2 variant than m-SPBs (*Figure 4ab*). In the vast majority of yeast cells, the b-SPB is the SPB inherited from the previous mitosis (old SPB), whereas the m-SPB is newly synthesized (*Pereira et al., 2001*). Spc42-mCherry can differentiate SPBs by age since Spc42 proteins in old SPBs carry mature – and thus reliably brighter – mCherry compared with immature mCherry in new SPBs (*Lengefeld et al., 2017*). When we separated the cells that correctly oriented the old SPB towards the bud from those few ones (6%) that inverted the orientation of their SPBs, we noticed that the asymmetry of Kip2-G374A-3xfsGFP was much stronger in cells with correct orientation (*Figure 4a*).

To quantify the asymmetry, we calculated an asymmetry index for Kip2 distribution, which is defined as the difference between the Kip2-G374-3xsfGFP signal on the m-SPB and on the b-SPB, normalized by the total signal of b-SPB and m-SPB. This index reaches one when Kip2-G374A-3xsfGFP is fully asymmetrically distributed towards the bud and −1 when it is fully asymmetric towards the mother cell. In about 94% of the cells, the SPBs are correctly oriented: their b-SPB is the old one and the median asymmetry index value is 0.7 (*Figure 4c*). Strikingly, also in cells with inverted SPB orientation, Kip2-G374A-3xfsGFP was significantly biased towards the old SPB, the m-SPB in this case. In these cells, asymmetry was less pronounced (median asymmetry index value: −0.2) than in cells with correct orientation, suggesting that some spatial information biased the recruitment of Kip2 towards the b-SPB as well. In cells with correctly oriented SPBs, the strong bias of Kip2 recruitment towards the b-SPB would then result from the effects of both the age and location of SPBs. We concluded that both the age of the SPB and the vicinity to the bud-neck synergistically activate SPBs for Kip2 recruitment.

To test whether the difference in Kip2-G374A-3xsfGFP recruitment between the two SPBs has any functional significance, we asked whether it was reflected in differences in the distribution of Kip2-3xsfGFP between m- and b-microtubules. In the few cells that carry a cytoplasmic microtubule on both sides of the spindle, Kip2 levels were nearly two-fold higher at the plus-ends of b-microtubules than of m-microtubules (*Figure 4de*). The asymmetry index of Kip2-3xsfGFP on microtubule plus-ends in cells with cytoplasmic microtubules emanating from both SPBs showed the same pattern as for Kip2-G374A-3xsfGFP on SPBs (*Figure 4df*). In cells with correctly oriented SPBs, the Kip2-3xsfGFP distribution was strongly biased towards the tip of b-microtubules, generally emanating from the old SPB, whereas in cells with inverted SPBs, it showed a weaker bias towards m-microtubules. Note that the difference in Kip2-3xsfGFP levels between m- and b-microtubules is less pronounced in cells with cytoplasmic microtubules on both SPBs than that of Kip2-G374A-3xsfGFP in all preanaphase cells. This may be explained by the different cell populations: cells that carry microtubules on both SPBs might correspond to those preanaphase cells that recruit Kip2 more symmetrically between SPBs, possibly allowing the m-microtubules to grow and become visible (see below). These data further supported the idea that SPBs are the main determinants for Kip2 recruitment to microtubules.

## Biased microtubule growth and dynein distribution correlates with Kip2 recruitment

The asymmetry of Kip2 recruitment to SPBs is highly reminiscent of both the asymmetry of cytoplasmic microtubule length and of dynein delivery. Indeed, the b-SPB, which recruits Kip2 more actively, causes the emanating microtubules to carry more Kip2, and on average grows microtubules twice as long as those emanating from the m-SPB (*Figure 4g*). Likewise, the increased loading of Kip2 on b-microtubules correlates well with the localization of dynein, one of the known cargos of Kip2 (*Carvalho et al., 2004*; *Caudron et al., 2008*; *Markus et al., 2009*; *Figure 4h*). Supporting the idea that dynein localization depends on Kip2 distribution, dynein levels at microtubule plus-ends were strongly decreased in *kip2Δ* mutant cells (*Carvalho et al., 2004*; *Caudron et al., 2008*;

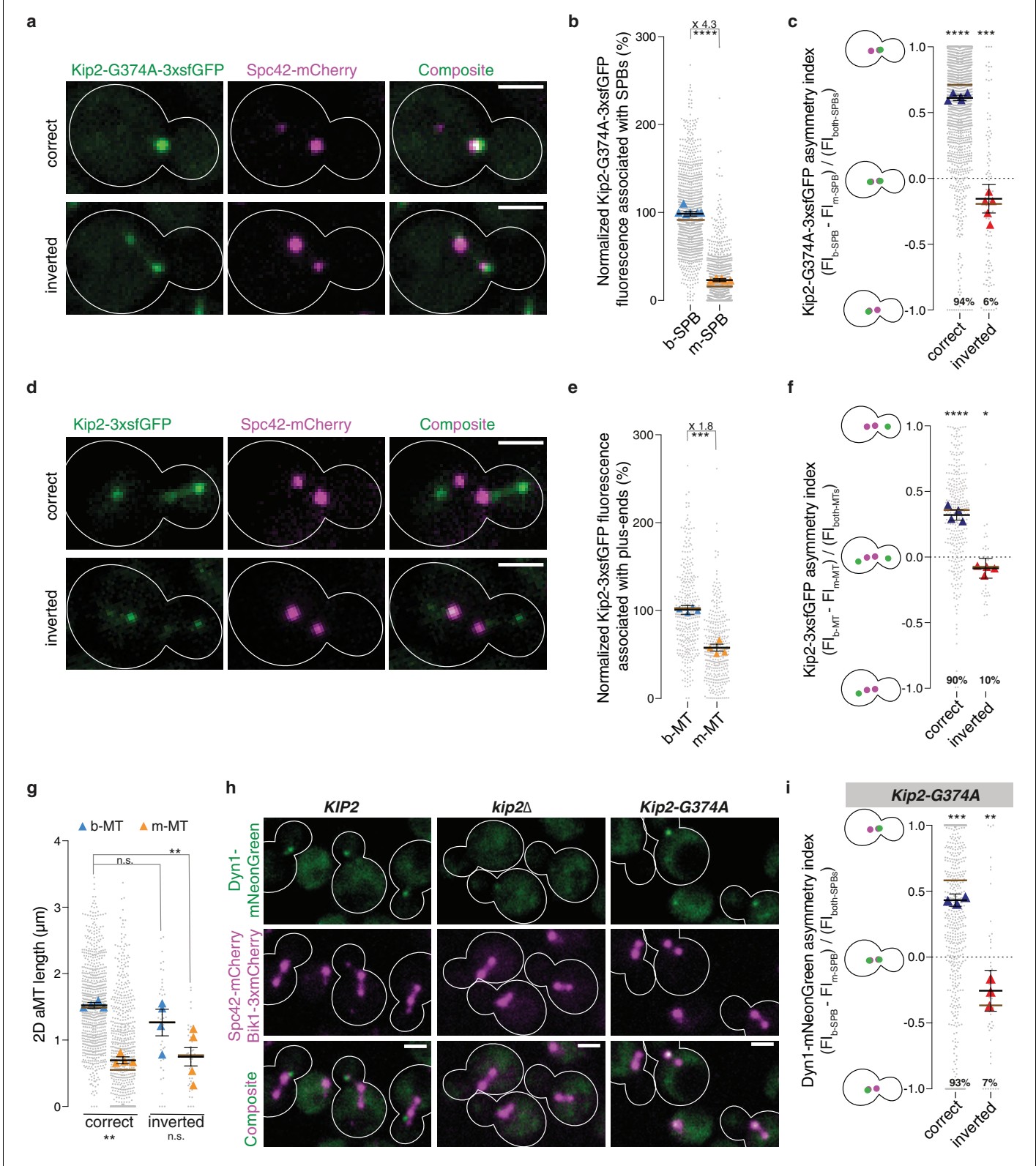

**Figure 4.** Biased microtubule growth and dynein distribution correlate with SPB dependent Kip2 recruitment. (**a**) Localization of Kip2-G374A-3xsfGFP (green) in preanaphase cells with correctly oriented (top) and inverted (bottom) SPBs. The orientation of the SPBs is visualized with Spc42-mCherry (magenta). (**b**) Relative Kip2-G374A-3xsfGFP fluorescence (%) associated with b- and m-SPBs in cells shown in (**a**) (999 cells from n > 3 independent clones or technical replicates). (**c**) Quantification of asymmetry index for Kip2-G374A-3xsfGFP distribution (fluorescence intensity: $(FI_{b\text{-}SPB} - FI_{m\text{-}SPB})$ /

*Figure 4 continued on next page*

**Figure 4 continued**

FI$_{both-SPBs}$) between SPBs in cells in (**a**) with correctly orientated (1407 cells) and inverted SPBs (97 cells). Statistical significances of difference from zero were tested with one-way ANOVA. (**d**) Localization of Kip2-3xsfGFP (green) in preanaphase cells carrying both b- and m- microtubules, with correctly oriented (top) and inverted (bottom) SPBs. (**e**) Normalized fluorescence (%) of Kip2-3xsfGFP associated with plus-ends of b- and m-microtubules in preanaphase cells in (**d**) (347 cells from n > 3 independent clones or technical replicates). (**f**) Quantification of asymmetry index for Kip2-3xsfGFP accumulation at m- and b-microtubule plus-ends (fluorescence intensity: (FI$_{b-MT}$ − FI$_{m-MT}$) / FI$_{both-MTs}$) in preanaphase cells in (**d**) with correctly orientated (314 cells) and inverted SPBs (33 cells). Statistical significances of difference from zero were tested with one-way ANOVA. (**g**) Measurements of two-dimensional (2D) b- and m- microtubule lengths (μm) in cells with correctly orientated (758 cells) and inverted SPBs (40 cells) using Kip2-3xsfGFP and Spc42-mCherry as microtubule plus- and minus-end markers, respectively. In case of no visible astral microtubule, the microtubule length was set to 0 μm. (**h**) Localization of Dyn1-mNeonGreen (green) in cells of indicated genotype. Spindles and aMTs are visualized with Spc42-mCherry and Bik1-3xmCherry (magenta). (**i**) Quantification of asymmetry index for Dyn1-mNeonGreen distribution (fluorescence intensity: (FI$_{b-SPB}$ − FI$_{m-SPB}$) / FI$_{both-SPBs}$) in *Kip2-G374A* cells with correctly orientated (641 cells) and inverted SPBs (47 cells). Note that in 13.4% (106 out of 794) of cells, no detectable Dyn1-mNeonGreen accumulated on preanaphase spindles; these cells were not included in the analysis. Statistical significances of difference from zero were tested with one-way ANOVA. For all panels, means with 95% confidence intervals are shown in black; median values shown as brown bar; data were acquired from at least three independent clones and technical replicates. Relative GFP fluorescence was obtained by normalizing to the mean GFP fluorescence associated with b-SPBs or plus-ends of b-microtubules, as indicated. The average value of each clone or technical replicate is plotted as triangle. ****p<0.0001, ***p<0.001, **p<0.01, *p<0.05, n.s., not significant. Statistical significances were calculated using one-way ANOVA (**b,c,e,f,g,i**). Source data for these panels are available in *Supplementary file 1*. Scale bars, 2 μm.
DOI: https://doi.org/10.7554/eLife.48627.013

---

*Markus et al., 2009* and *Figure 4h*), and dynein was restricted to SPBs in the *Kip2-G374A* mutant cells (*Figure 4h*). Furthermore, dynein distribution showed the same correlation with old SPBs as Kip2, being most asymmetric in cells with correctly oriented SPBs, and slightly asymmetric towards the old SPB when SPBs were inverted (*Figure 4i*). Thus, SPB-dependent recruitment of Kip2 to microtubules might be one of the mechanisms through which yeast cells promote the growth of b-microtubules and bias dynein distribution towards them.

## Bfa1 and Bub2 promote Kip2 run initiation from bud-directed SPBs

We next sought to determine how cells restrict Kip2 localization to b-SPBs and b-microtubules and to develop perturbations affecting this process. Since Kip2 recruitment to SPBs was at least in part dependent on SPB age, we wondered whether mechanisms specifying the age of SPBs contribute to the preference of Kip2 for the old SPB. The SPB-associated proteins Bub2 and Bfa1 function together as a bipartite GTPase-activating protein (GAP) complex for the GTPase Tem1 and partly independently of each other in SPB specification. In early preanaphase, they are recruited to the old SPB upon its specification by the Spindle Pole Inheritance Network (SPIN). When the SPB moves to the vicinity of the bud neck later in preanaphase, the levels are further enhanced by the position of the SPB. As a consequence, they are most strongly biased to the old SPB when this SPB is the b-SPB compared to when it is the m-SPB (*Lengefeld et al., 2017*). In anaphase cells, Bub2 and Bfa1 accumulate to high levels on the b-SPB as it enters the bud, irrespective of the SPB's age (*Bardin et al., 2000*; *Pereira et al., 2000*). Therefore, the pattern of Bub2 and Bfa1 distribution is highly reminiscent of that of Kip2.

Thus, we next asked whether Bub2 and Bfa1 influence the recruitment of Kip2 to SPBs. We quantified Kip2-G374A-3xsfGFP localization to SPBs in the *bfa1Δ* and *bub2Δ* single and *bfa1Δ bub2Δ* double mutant cells. Either of these single or double deletions lowered the levels of Kip2-G374A-3xsfGFP on the b-SPB almost by half, while the Kip2-G374A-3xsfGFP signal on the m-SPB was unchanged (*Figure 5ab*). The deletions had the known effect of making the SPB orientation more random (*Lengefeld et al., 2017* and *Figure 5c*). The asymmetry index indicated that cells with properly oriented and inverted SPBs now behaved very similarly: compared to wild type cells, the asymmetry of ATPase deficient Kip2 was strongly reduced on properly oriented SPBs and fully erased on inverted SPBs (*Figure 5c*). Importantly, *bfa1* and *bub2* deletions did not affect the pattern of Kip2 distribution along microtubule shafts, which remained flat, peaking only at the plus-ends, independent of microtubule length (*Figure 5d* and *Figure 5—figure supplement 1*). However, the intensity on the lattice was reduced. This was reflected in the median estimates provided by our in silico model. The median in-rate constant (recruitment rate at the minus-end, k$_{in}$) was 12 times lower in the *bub2Δ bfa1Δ* double mutant than in wild type cells (*Figure 5e*, p=0.0008 when assuming a Kip2 concentration of ≥ 35 nM, *Figure 2*; *Figure 2—figure supplement 3abc*, see Appendix1 for details),

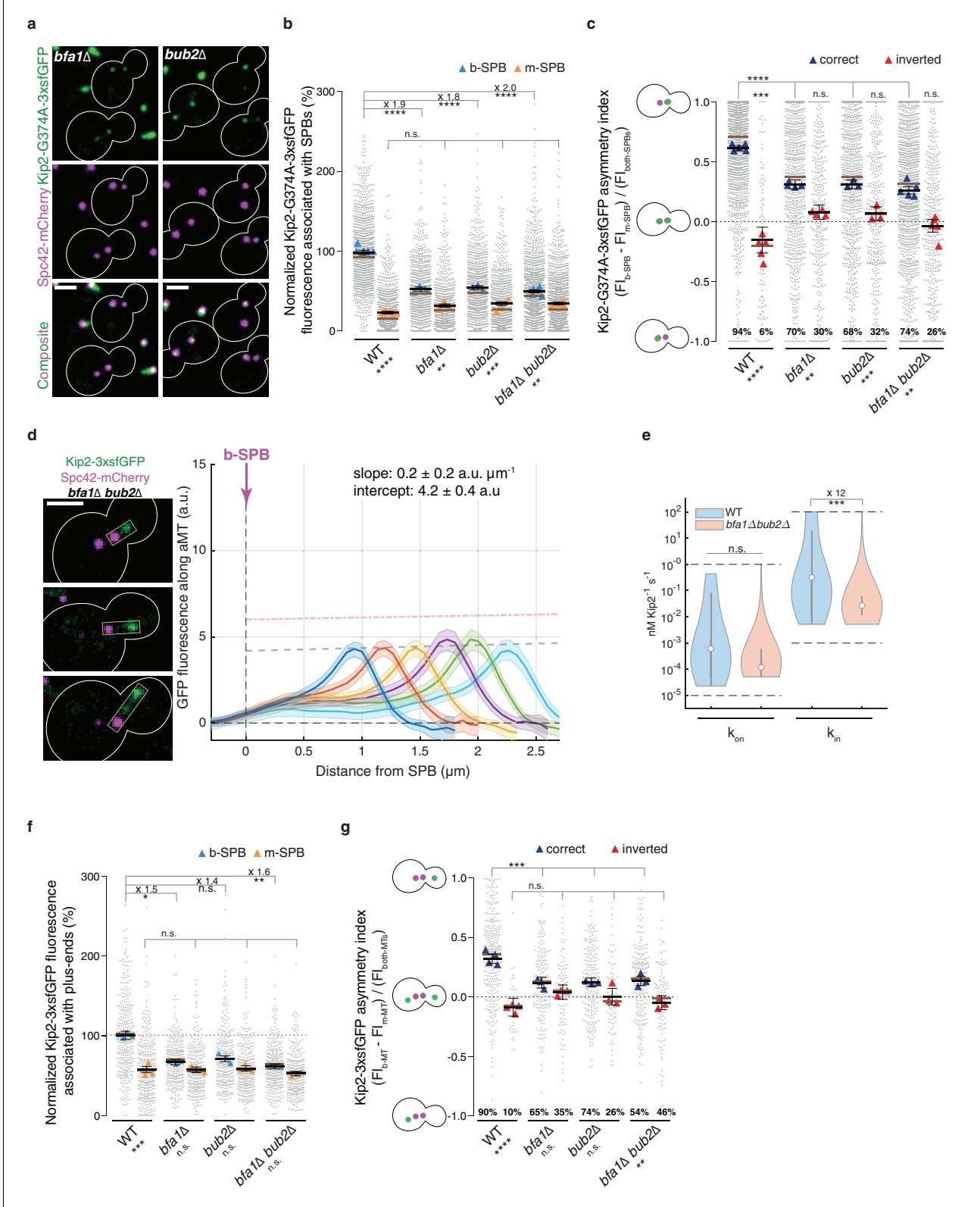

**Figure 5.** Bfa1 and Bub2 promote Kip2 run initiation from bud-directed SPBs. (**a**) Representative images of Kip2-G374A-3xsfGFP (green) in preanaphase cells of indicated genotype. Spindles are visualized with Spc42-mCherry (magenta). See *Figure 3c* for control. (**b**) Normalized Kip2-G374A-3xsfGFP fluorescence (%) associated with b- (blue) and m-SPBs (yellow) in cells shown in (**a**) (n > 3 independent clones or technical replicates,>701 cells per genotype). (**c**) Quantification of asymmetry index for Kip2-G374A-3xsfGFP distribution (fluorescence intensity: (FI$_{b-SPB}$ – FI$_{m-SPB}$) / FI$_{both-SPBs}$) between

*Figure 5 continued on next page*

*Figure 5 continued*

SPBs in cells in (**b**) with correctly orientated (blue) and inverted (red) SPBs. For cells with inverted SPBs, statistical significances of difference from zero were tested with one-way ANOVA. (**d**) Representative images (left) and quantifications (right) of fluorescence intensities (a.u.) from endogenous Kip2-3xsfGFP along preanaphase aMTs (boxed areas) in *bfa1Δbub2Δ* cells. The graph format is the same as in *Figure 1*. $45 \leq n \leq 99$ per bin. The pink dashed line denotes the weighted linear regressions for the mean GFP fluorescence on plus-ends in wild-type cells. See *Figure 5—figure supplement 1* for more details. (**e**) In silico likelihood of on rate constant $k_{on}$ and in rate constant $k_{in}$ estimated from model fits to Kip2-3xsfGFP distribution in wt and *bfa1Δbub2Δ* cells. Statistical significance for $k_{in,bfa1\Delta bub2\Delta} < k_{in,wt}$ (***, p=8•$10^{-4}$, was determined by sampling from the likelihood for $[\text{Kip2}]_{total} \geq 35\,\text{nM}$ (see Appendix 1), difference in median as indicated. Graph as in *Figure 2d*. (**f**) Normalized Kip2-3xsfGFP fluorescence (%) associated with microtubule plus-ends in preanaphase cells carrying both b- (blue) and m- (yellow) microtubules (n > 3 independent clones or technical replicates,>275 cells per genotype). See *Figure 5—figure supplement 1a* for representative images. (**g**) Quantification of asymmetry index for Kip2-3xsfGFP accumulation at m- and b-microtubule plus-ends (fluorescence intensity: $(\text{FI}_{b-MT} - \text{FI}_{m-MT}) / \text{FI}_{both-MTs}$) in preanaphase cells for data in (**f**) with correctly orientated (blue) and inverted (red) SPBs. Graphs and statistical analysis are as in *Figure 4 (b,c,e,f,g and i)* unless otherwise indicated. Statistical significances within each genotype are marked at the bottom. Scale bars, 2 µm. See also *Figure 5—figure supplement 1*.
DOI: https://doi.org/10.7554/eLife.48627.014

The following figure supplement is available for figure 5:

**Figure supplement 1.** Kip2 localization patterns along microtubules in cells with both b- and m- microtubules.
DOI: https://doi.org/10.7554/eLife.48627.015

while the median on-rate constant ($k_{on}$) was not significantly reduced. Together, these data indicated that in cells lacking Bub2 and Bfa1, the initiation of Kip2 runs was still restricted to SPBs but was reduced on the b-SPB to the level observed on the m-SPB. Thus, we investigated whether the reduction of Kip2 recruitment by b-SPBs was reflected by the levels of Kip2-3xsfGFP on microtubules. Indeed, in all *bfa1Δ* and *bub2Δ* single and double mutant cells Kip2-3xsfGFP levels dropped at the plus-ends of b-microtubules to nearly m-microtubule levels, while remaining unchanged at the plus-ends of m-microtubules, compared to wild type cells (*Figure 5f*). Consistently, Kip2-3xsfGFP levels at the plus-ends of microtubules in cells with both cytoplasmic microtubules were now more symmetric (*Figure 5g* and *Figure 5—figure supplement 1*). Hence, Bub2 and Bfa1, which respond synergistically to historical and spatial cues to accumulate specifically on the b-SPB, contribute together to Kip2 recruitment on the b-SPB and to the enhanced rate of Kip2 run initiation on b-microtubules.

## Phosphorylation of its N-terminus prevents Kip2 from landing along microtubules

Besides SPB recruitment, in silico modeling indicated that Kip2 distribution also depended on forbidding landing and run initiation at random places on microtubules. Interestingly, Kip2's N-terminus is heavily phosphorylated in vivo in a GSK3-, Cdk1-, and Dbf2/20-dependent manner (*Drechsler et al., 2015*) (*Figure 6a*); the kinases Dbf2 and Dbf20 function in the yeast Hippo pathway (*Hergovich et al., 2006*). GSK3-dependent phosphorylation is primed through phosphorylation of serine 63, which is conserved across Kip2 orthologues in fungi (*Figure 6—figure supplement 1*) and falls into a consensus site for the mitotic kinases Cdk1 and Dbf2/20. Furthermore, phosphorylation of the N-terminus of Kip2 inhibits Kip2 binding to microtubules (*Drechsler et al., 2015*). As reported (*Drechsler et al., 2015*), mutation of S63 to alanine (*KIP2-S63A*) largely reduced Kip2 phosphorylation (*Figure 6b*). Interestingly, both SPBs recruited more of the hypo-phosphorylated, ATPase deficient protein Kip2-S63A-G374A-3xsfGFP (*Figure 6cd*) but this recruitment was still asymmetric between SPBs, except in cells with inverted SPB orientation (*Figure 6e*). Thus, preventing Kip2 phosphorylation did not affect much the role of the SPBs in Kip2 recruitment. In contrast, the distribution profile of Kip2-S63A-3xsfGFP along microtubules deviated strongly from that of Kip2-3xsfGFP and resembled more that of Kip3 (*Figure 1a*): The levels of Kip2-S63A-3xsfGFP linearly increased from the minus- to the plus-ends of the microtubules, and plus-end levels increased with microtubule length (*Figure 6f*, *Video 3*). We estimated the effect of the S63A mutation on the kinetic parameters driving Kip2 distribution using our in silico model as above. These estimates indicate that the median on-rate constant of Kip2-S63A ($k_{on}$) is significantly higher than for the wild type protein (p<2•$10^{-4}$ when assuming a Kip2 concentration of $\geq 35\,\text{nM}$, *Figure 2* and *Figure 2—figure supplement 3abd*, see Supplemental information for details), while the median in-rate constant ($k_{in}$, minus-end recruitment) is not significantly changed (*Figure 6g*). These estimates indicate that phosphorylation of Kip2's N-terminus primarily inhibits Kip2 from landing on microtubules. Consistent

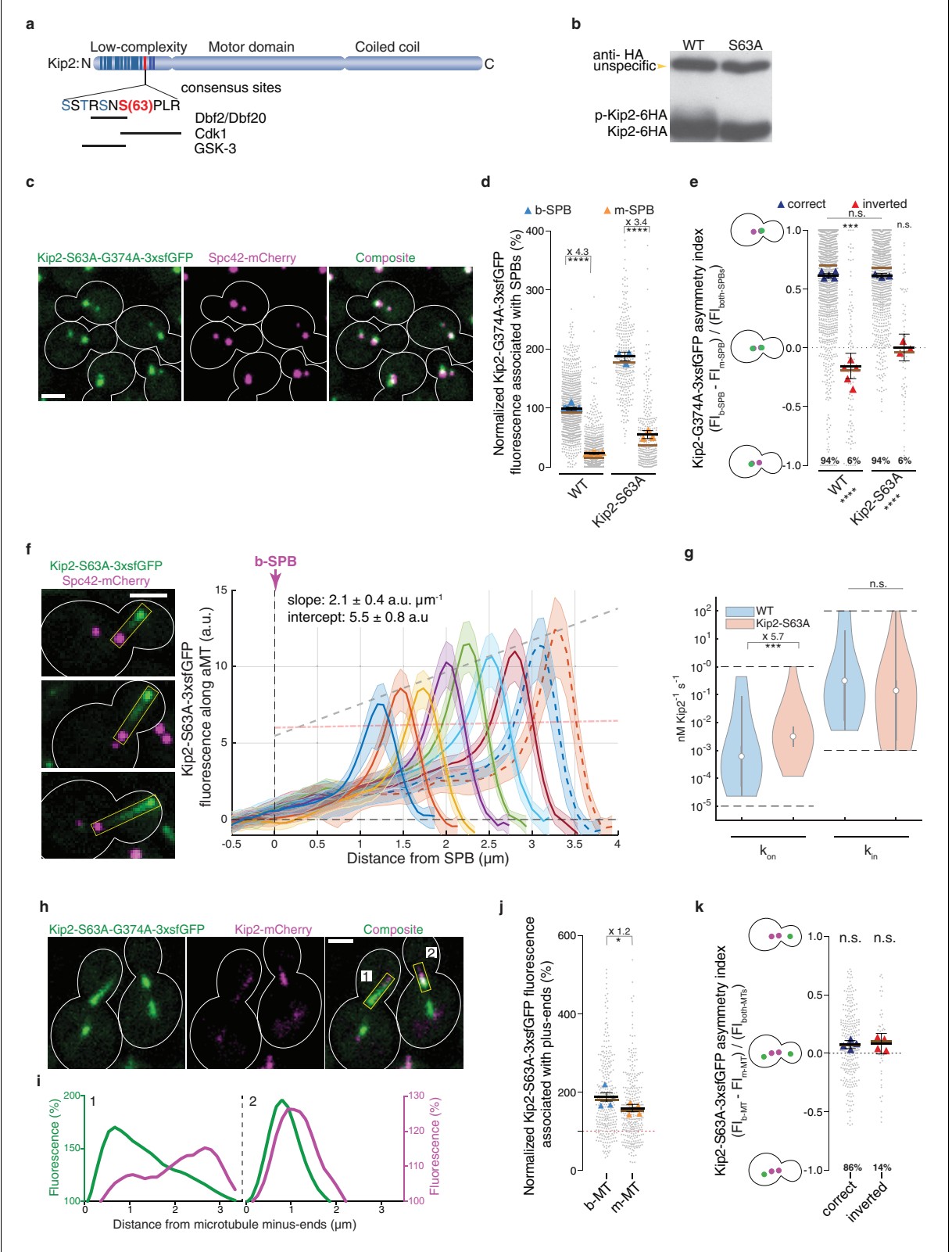

**Figure 6.** Phosphorylation of its N-terminus prevents Kip2 from landing along microtubules. (a) Scheme of Kip2 protein and its disordered N-terminus with residue serine 63 (red) and kinase consensus sites (blue). (b) Western blot analysis of endogenously expressed Kip2-6HA and Kip2-S63A-6HA. Lysates were prepared from cycling cells of indicated genotype. (c) Representative images of Kip2-S63A-G374A-3xsfGFP (green) in preanaphase cells. Spindles are visualized with Spc42-mCherry (magenta). See *Figure 3b* for control. (d) Normalized Kip2-S63A-G374A-3xsfGFP fluorescence (%)

*Figure 6 continued on next page*

Figure 6 continued

associated with b- (blue) and m-SPBs (yellow) in cells shown in (**c**) (n > 3 independent clones or technical replicates,>314 cells per genotype). (**e**) Quantification of asymmetry index for Kip2-S63A-G374A-3xsfGFP distribution (fluorescence intensity: ($FI_{b-SPB} - FI_{m-SPB}$) / $FI_{both-SPBs}$) between SPBs for data in (**d**) with correctly orientated (blue) and inverted (red) SPBs. (**f**) Representative images (left) and quantifications (right) of fluorescence intensities (a.u.) from endogenous Kip2-S63A-3xsfGFP along preanaphase aMTs (boxed areas). The graph format is the same as those in *Figure 1*. $29 \leq n \leq 55$ per bin. The pink dashed line denotes the weighted linear regression for the mean GFP fluorescence on plus-ends in wild-type cells. See *Figure 5— figure supplement 1* for more details. (**g**) In silico likelihood of on rate constant $k_{on}$ and in rate constant $k_{in}$ estimated from model fits to Kip2-3xsfGFP data in wt and Kip2-S63A-3xsfGFP cells. Statistical significance for $k_{on,S63A} > k_{on,wt}$ (***, p<$2 \cdot 10^{-4}$) was determined by sampling from the likelihood for $[Kip2]_{total} \geq 35\,nM$ (see Appendix 1), difference in median as indicated. Graph as in *Figure 2d*. (**h**) Representative images of preanaphase heterozygous diploid cells expressing the endogenous ATPase deficient protein Kip2-S63A-G374A-3xsfGFP (green) and the wild type protein Kip2-mCherry (magenta). (**i**) Line scan analysis along the numbered microtubules shown in (**h**). As in *Figure 3b*, GFP (green) and mCherry (magenta) fluorescence intensities were normalized to their background levels, respectively. (**j**) Normalized Kip2-S63A-3xsfGFP fluorescence (%) associated with microtubule plus-ends in preanaphase cells carrying both b- (blue) and m- microtubules (yellow) (305 cells from n > 3 independent clones). Values were normalized to Kip2-3xsfGFP fluorescence associated with b-microtubules in wild-type cells, denoted as red dotted line in the graph. See *Figure 5—figure supplement 1a* for representative images. (**k**) Quantification of asymmetry index for Kip2-S63A-3xsfGFP accumulation on microtubule plus-ends (fluorescence intensity: ($FI_{b-MT} - FI_{m-MT}$) / $FI_{both-MTs}$) for data in (**j**) with correctly orientated (blue) and inverted (red) SPBs. Graphs and statistical analysis are like those in *Figure 4 (b,c,e,f,g and i)*. Statistical significances within each genotype are marked at the bottom of graphs. Source data for these panels are available in *Supplementary file 1*. Scale bars, 2 µm. See also *Video 3*, *Figure 5—figure supplement 1* and *Figure 6—figure supplement 1*.

DOI: https://doi.org/10.7554/eLife.48627.016

The following figure supplement is available for figure 6:

**Figure supplement 1.** Conservation of Kip2's low complexity N-terminus.

DOI: https://doi.org/10.7554/eLife.48627.017

with this hypothesis, the hypo-phosphorylated, ATPase-deficient Kip2-S63A-G374A-3xsfGFP protein decorated both microtubule shafts and SPBs in heterozygous diploid cells co-expressing Kip2-mCherry (*Figure 6hi*). Furthermore, the Kip2-S63A mutation largely bypassed the differential control of Kip2 recruitment to microtubules exerted by SPBs. We reasoned that the approximately 6-fold increase of the on-rate constant, which was still around 40-fold lower than the in-rate constant, had such a profound effect because Kip2 recruitment rates are proportional to the number of recruitment sites, and lattice sites are more abundant than sites at the minus-end. The levels of Kip2-S63A-3xsfGFP on plus-ends increased on both m- and b-microtubules, and the level differences between them were substantially reduced compared to wild type (*Figure 6j* and *Figure 4e*). Moreover, the asymmetry of hypo-phosphorylated Kip2-S63A-3xsfGFP levels between m- and b-microtubule plus-ends was reduced and no-longer depended on whether the spindle was properly oriented or not (*Figure 6k* and *Figure 4f*). Therefore, preventing Kip2 landing along microtubules is a pre-requisite for the control of Kip2 recruitment by SPBs and Kip2 phosphorylation is key in this process.

## SPB dependent recruitment of Kip2 specifies microtubule length and dynein distribution

While targeted Kip2 recruitment at SPBs (via Bub2/Bfa1) and prevention of random lattice binding (via Kip2 phosphorylation) ensured that Kip2 distributions were biased towards the plus-ends of b-microtubules independently of microtubule length, we wondered what the functional consequences of such Kip2 distribution are. Therefore, we asked whether the *bub2Δ*, *bfa1Δ* and *KIP2-S63A* mutations, which mitigate the control of Kip2, affected the length of m- and b-microtubules and the delivery of dynein to their plus-ends. We used the yeast homolog of CLIP-170, Bik1, to label microtubule plus-ends, as reported (*Lengefeld et al., 2018*; *Stangier et al., 2018*), and Spc72 as the minus-end marker. Bik1 and Spc72 were fused to three and one copy of GFP, respectively, at their endogenous loci. The Bik1-3xGFP reporter localized most strongly to microtubule plus-ends but decorated microtubule shafts and minus-ends as well (*Figure 7a* and *Figure 7—figure supplement 1a*), allowing accurate measurement of three-dimensional (3D) microtubule length for microtubules longer than 0.67 µm (see Material and methods). As already reported (*Lengefeld et al., 2018*), most preanaphase cells did not form a measurable microtubule on their m-SPB. This remained true in the *KIP2-S63A*, *bub1Δ*, *bfa1Δ* single and the *bub2Δ bfa1Δ* double mutant strains (*Figure 7—figure supplement 1b*), indicating that neither Kip2 phosphorylation nor Bub2 and Bfa1 control the differential rate of microtubule nucleation between m- and b-SPBs by themselves.

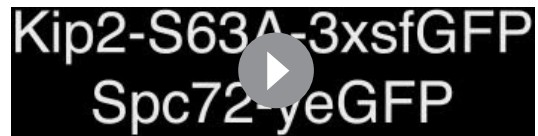

**Video 3.** A representative time-lapse movie shows that unlike wildtype molecules, Kip2-S63A-3xsfGFP molecules accumulate along the cytoplasmic microtubule shaft length-dependently. Also, these speckles move towards and reach the plus-end. The movie consists of 80 frames that were taken every 1.07 s and the frame speed is sped up by 3-fold for better visualization. Scale bar, 2 µm.

DOI: https://doi.org/10.7554/eLife.48627.018

In the wild type cells that formed both m- and b-microtubules, the b-microtubules were on average longer than the m-microtubules (*Figure 4g* and *Figure 7b*; *Lengefeld et al., 2018*). Kip2 hypo-phosphorylation (*KIP2-S63A* mutant cells) caused both m- and b-microtubules to become longer on average compared to wild type cells (*Figure 7ab*), in agreement with these microtubules carrying higher levels of the microtubule polymerizing kinesin Kip2 at their plus-ends (*Figure 6j* and *Figure 5—figure supplement 1*). In these mutant cells, the m- and b-microtubules were no-longer significantly different in length. Inactivation of Bfa1, Bub2, or both also eliminated the length difference between m- and b-microtubules, but in this case by reducing the average length of b- to that of m-microtubules (*Figure 7ab*). This fully agrees with reduced Kip2 levels on both the b-SPB and b-microtubules compared to wild type, but unchanged levels on m-SPBs and m-microtubules (*Figure 5f*). Consistent with these effects, the *KIP2-S63A, bfa1Δ, bub2Δ,* and *bfa1Δ bub2Δ* mutations randomized dynein distribution in cells carrying both b- and m- microtubules (*Figure 7cd*). Unlike in wild type cells, Dyn1-mNeonGreen localized indistinguishably to the tip of both b- and m-microtubules in all these mutant cells. The amount of Dynein molecules appears to be limiting in these cells as they are redistributed between the two microtubule plue-ends in *Kip2-S63A* cells (*Figure 7—figure supplement 2*). We therefore concluded that restricting the initiation of Kip2 runs to the b-SPBs functioned as a molecular determinant for specifying the length and function of b-microtubules in vivo.

## Discussion

Here, we identify a remote-control mechanism for patterning microtubule organization and function at the subcellular level in yeast and demonstrate that this mechanism relies on a dedicated mode of kinesin regulation (*Figure 7e*). We provide evidence that the messenger-kinesin Kip2, a member of the kinesin-7 family, makes the growth of microtubules and their role in cargo delivery dependent on the MTOC they emanate from. This process relies on two main conditions: the cytoplasmic pool of the kinesin is inhibited in its ability to bind microtubules at random positions, and at least one MTOC provides activities promoting the recruitment and reactivation of the kinesin locally, at the minus-end of the microtubule. Together, these two conditions ensure that the kinesin accumulates on only a subset of microtubules. The kinesin in turn determines the dynamic properties of these microtubules: Kip2 stabilizes the targeted microtubules (*Hibbel et al., 2015*), enables them to grow longer and to reach into the bud, where they deliver specific cargos such as dynein (*Moore et al., 2009*). As a minus-end directed microtubule motor itself, dynein contributes further to defining the functional properties of the microtubules. Releasing the specificity of the SPB or the cytoplasmic inhibition of the kinesin equalizes the distribution of the kinesin and its cargoes, and prevents microtubule differentiation. Thus, at least in budding yeast, the remote-control of microtubule plus-ends by the SPBs is a major mechanism for establishing spindle asymmetry and differentiating the behavior of selected microtubules.

Interestingly, modification of the kinesin rather than of microtubules prevented cytoplasmic Kip2 from landing at random locations along microtubules. Landing-inhibition was achieved through phosphorylation, probably the simplest mechanism possible. Remarkably, phosphorylation involves a domain of Kip2 with many potential phosphorylation sites, most of which are targeted by yeast GSK3-related kinases (*Drechsler et al., 2015*). Once initiated by the priming kinase, Gsk3 can very efficiently maintain Kip2 in a highly phosphorylated state. Where the priming event of phosphorylation takes place is unknown at this stage. It could involve a cytoplasmic kinase, such as cytoplasmic Cdk1 and Dbf2 activities (*Drechsler et al., 2015*), or a kinase localized to the microtubule plus-end, such as Cdk1, which accumulates at the tip of b-microtubules during preanaphase (*Maekawa et al., 2003*; *Maekawa and Schiebel, 2004*). Interestingly, preventing Kip2 phosphorylation did not slow

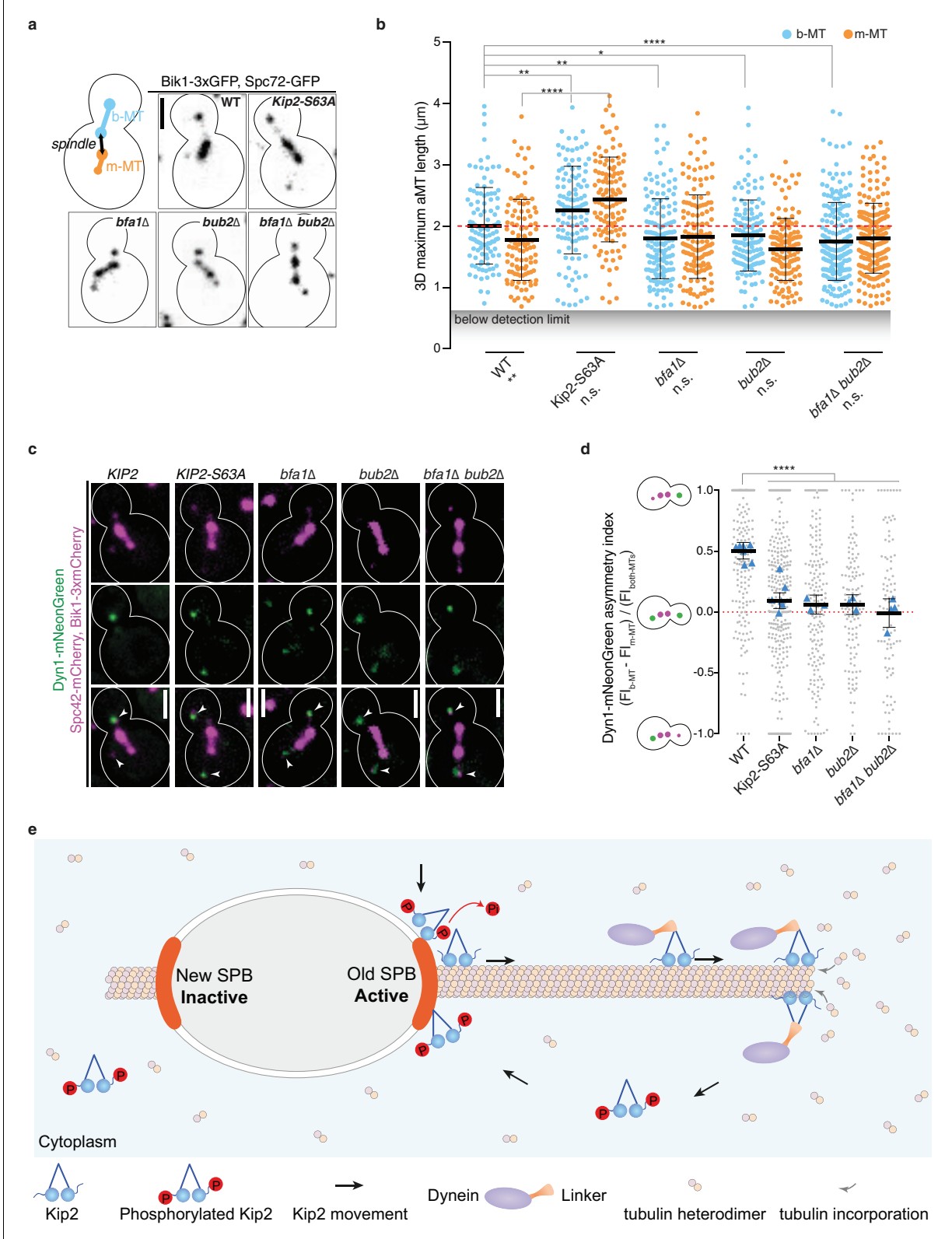

**Figure 7.** SPB dependent recruitment of Kip2 specifies aMT size and dynein distribution. (**a**) Illustration and representative images of preanaphase cells of indicated genotype carrying both b- and m-microtubules using Bik1-3xGFP and Spc72-GFP as microtubule plus- and minus-end markers, respectively. Scale bars, 2 μm. (**b**) Quantification of the three-dimensional (3D) average maximum length (μm) of b- (blue) and m-microtubules (yellow) over the recording time (85.6 s, see Materials and methods) from the cells shown in (**a**). Means with 95% confidence intervals are shown in black,

*Figure 7 continued on next page*

*Figure 7 continued*

n > 127 cells per genotype. ****p<0.0001, ***p<0.001, **p<0.01, *p<0.05, n.s., not significant. Statistical significances were calculated using two-tailed Student's t-test. Source data are available in *Supplementary file 1*. (c) Localization of Dyn1-mNeonGreen (green) in preanaphase cells of indicated genotype carrying both aMTs. Spindles and aMTs are visualized with Spc42-mCherry and Bik1-3xmCherry (magenta). Scale bars, 2 μm. (d) Quantification of asymmetry index (fluorescence intensity: $(FI_{b-MT} - FI_{m-MT}) / FI_{both-MTs}$) of Dyn1-mNeonGreen accumulation on microtubule plus-ends in cells shown in (c) (>94 cells per genotype). Means with 95% confidence intervals are shown in black; data were acquired from at least three independent clones or technical replicates. The average value of each clone or technical replicate is plotted as triangle. ****p<0.0001, Statistical significances were calculated using one-way ANOVA. Source data for this panel are available in *Supplementary file 1*. (e) Illustration of the proposed remote-control mechanism (see main text for details). See also *Figure 7—figure supplement 1* and *Figure 7—figure supplement 2*.
DOI: https://doi.org/10.7554/eLife.48627.019

The following figure supplements are available for figure 7:

**Figure supplement 1.** Astral microtubule organization in preanaphase yeast cells.
DOI: https://doi.org/10.7554/eLife.48627.020

**Figure supplement 2.** Pairwise Dyn1-mNeonGreen distribution between b- and m-microtubules in control and Kip2-S63A cells.
DOI: https://doi.org/10.7554/eLife.48627.021

down its release from microtubule tips ($k_{out}$), indicating that even if Kip2 phosphorylation took place at microtubule tips, it is not the determining step for releasing the kinesin. Thus, Kip2 phosphorylation specifically prevents reloading of cytoplasmic Kip2 to random places on microtubules and thereby ensures the supremacy of SPBs in the control of Kip2 recruitment.

As a consequence, Kip2 recruitment activity at SPBs most likely requires two components. First, a high-affinity or many low-affinity binding sites for phosphorylated Kip2 must be present at the MTOC to ensure its recruitment against competing binding sites on microtubules: even if phosphorylated Kip2 binds them only poorly, the surface of the microtubules in the cell and the number of tubulin dimers that they expose is orders of magnitude greater than the size of the microtubule minus-end. A function of MTOCs in recruiting microtubule-associated proteins may explain the comparatively large size of the outer plaque of SPBs for nucleating only very few cytoplasmic microtubules, and may underlie the gel-like structure and size of the peri-centriolar material in animal centrosomes (*Woodruff et al., 2017*). Second, at SPBs the kinesin must be released of the inhibition that prevents its binding to the microtubule lattice. Indeed, the ability to bind the lattice is consubstantial to the ability of these motor proteins to walk processively along the microtubules. This suggests that an as yet unidentified phosphatase locally activates Kip2 at SPBs. Thus, our data suggest that understanding to which extent and how MTOCs differentiate themselves from each other to regulate processes such as the recruitment of messenger kinesins will be paramount to understanding many aspects of microtubule patterning in cells, particularly during mitosis.

We suggest that the remote-control model of microtubule behavior by MTOCs as identified here is conserved beyond yeast and might even provide a useful perspective for solving open issues of how the tubulin code is established and maintained in vivo. In interphase and post-mitotic cells, post-translational modifications of tubulin were identified in the microtubule lattice and proposed to play a key role in determining the distinctive dynamics and functional properties of individual microtubules. These mechanisms are collectively referred to as 'tubulin code' (*Gadadhar et al., 2017*). Tubulin modifications directly or indirectly affect the stability of the microtubule, the recruitment of MAPs and specific +TIPs, and the binding and motor activity of kinesins and dynein (*Gadadhar et al., 2017*; *Song and Brady, 2015*). In all cases, however, how cells control the recruitment of tubulin-modifying enzymes to specific microtubules remains largely unclear (*Roll-Mecak, 2019*).

We propose that the recruitment of specific kinesins by MTOCs could be a mechanism for controlling the recruitment of tubulin modifiers. Indeed, recent data in mammalian cells indicate that kinesins control at least some microtubule modifications such as acetylation (*Tang et al., 2018*; *Wang et al., 2019*). In addition, microtubule longevity is a prerequisite for the acquisition of many tubulin modifications (*Roll-Mecak, 2019*). Therefore, microtubule-stabilizing messenger kinesins such as Kip2 are excellent candidates for modulating the propensity of microtubules to acquire specific modifications. In that regard, we note that MTOC asymmetry can precede asymmetric tubulin modification and microtubule spatial orientation in the meiotic spindle (*Wu et al., 2018*). Finally, it is tempting to speculate that at least some tubulin modifiers are actually kinesin cargos.

To establish that our remote-control model applies in general, we would rely on evidence that other cell types also exhibit MTOC-dependent control of microtubule behavior and on evidence that this control is achieved by messenger kinesins. Although much work is needed to make any firm conclusion, both conditions may be fulfilled. Indeed, yeast cells are not the only asymmetrically dividing cells that form asters of different sizes during mitosis. The one cell embryo of *C. elegans*, *Drosophila* neuroblasts and male germline stem cells show similar asymmetries (*Januschke and Gonzalez, 2010*; *Yamashita et al., 2007*; *Grill et al., 2001*). Furthermore, centrosomes segregate non-randomly and as a function of their age in many stem cells, from fruit fly to the mouse (*Lengefeld and Barral, 2018*). Thus, in many cell-types the centrosomes seem to be able to control the plus-end dynamics and functions of the microtubules that they nucleate, depending on centrosome age. Furthermore, at least a few kinesins feature the critical ability to lose the ability to bind and walk along microtubules when phosphorylated in organisms as distant from yeast as metazoans (*Kevenaar et al., 2016*; *Kelliher et al., 2018*; *Drerup et al., 2016*); this supports the notion that kinesins can function as messengers between MTOCs and microtubule plus ends in many eukaryotes.

Finally, it is clear that not all kinesins are regulated by MTOCs. As we show for preanaphase yeast cells, cytoplasmic microtubules are decorated in a SPB-independent but length-dependent manner by a microtubule-destabilizing kinesin-8 (Kip3), and concomitantly in a SPB-dependent but length-independent manner by a microtubule-stabilizing kinesin-7 (Kip2). The fission yeast *S. pombe* homologs of Kip2 (Tea2) and Kip3 (Klp5/6) were recently shown to exhibit a similar pattern of kinesin accumulation at microtubule plus-ends (*Meadows et al., 2018*). The two distinct but co-existing modes of kinesin recruitment likely underlies microtubule stability control by kinesin competition at microtubule plus ends (*Meadows et al., 2018*), leading to a complex balance of regulation on different microtubules. It will be interesting to dissect how this balance might contribute to precisely defining the length of individual microtubules. Thus, the control of kinesin distribution emerges here as a code in itself that has distinct and predictable effects on the function and dynamics of the underlying microtubules, as well as on the patterning of cellular microtubule organization.

## Materials and methods

### Yeast strains

Yeast strains used in this study are listed in *Supplementary file 2*. All strains are isogenic to S288C. Fluorescent or HA-tagged proteins were tagged at endogenous loci (*Knop et al., 1999*). All gene deletions were created using the PCR-based integration system (*Janke et al., 2004*) and gene deletions were verified by PCR analysis. Specific Kip2 mutations were introduced on a pRS314-Kip2-3xsfGFP:KanMX plasmid or a pRS304-Kip2 plasmid via site-directed mutagenesis (pfu-Turbo, Stratagene). KIP2 locus was then amplified and integrated in a *kip2Δ* strain and the correct integration was verified by PCR and sequencing.

### Media and growth conditions

Cells were cultured in YEPD (yeast extract peptone, 2% dextrose) for collecting western blotting samples. For live cell imaging, overnight cultures in SC (synthetic medium, 2% dextrose) were diluted to $OD_{600}$ 0.15 and cultivated for four more hours before being placed on an SC-medium agar patch for microscopy imaging.

### Fluorescence microscopy

A Nipkow spinning disk (Carl Zeiss) equipped with an incubator for temperature was employed. Time-lapse movies were acquired using a back-illuminated EM-CCD camera Evolve 512 (Photometrics, Inc) mounted on the spinning disk microscope with a motorized piezo stage (ASI MS-2000) and $100 \times 1.46$ NA alpha Plan Apochromat oil immersion objective, driven by Metamorph based software VisiVIEW (Visitron Systems). 17 Z-section images separated by 0.24 µm increments were captured with the exposure time of 30 ms each, the whole stack took 1.07 s. For imaging aMT dynamics, 80 continuous repetitions were taken. For imaging strains with both GFP and mCherry signals, the GFP channel was always set to 30 ms exposure time and the mCherry channel to 50 ms exposure time. For diploid cells expressing Kip2-mCherry, the exposure time for the mCherry

channel was set at 100 ms. For imaging strains expressing Dyn1-mNeonGreen, Spc42-mCherry, and Bik1-3xmCherry, both the mNeonGreen channel and the mCherry channel were exposed for 200 ms each. Images in figures represent sum fluorescence intensities across Z-projections. Scale bars represent 2 μm.

## Image and data analysis

Preanaphase cells were collected based on the shape of cells and the size of spindles. For analyzing the profiles of GFP fusion proteins along astral microtubules (aMTs), the sum intensity projection of the images was used. A 5-pixel (666.7 nm) width line was used to scan astral microtubules from plus-ends toward SPBs both in the GFP and the mCherry channels using Fiji (*Schindelin et al., 2012*), and exported to CSV files. These profiles were then aggregated for further analysis using MATLAB (R2018a, Mathworks), and peak detection for the GFP and mCherry signals was performed, respectively. Profile length was defined as the peak-to-peak distance, and the profiles were then binned into length bins as detailed in the figure legends.

Normalization between profile data acquired on different dates was performed by acquiring data from the same wild-type (wt) strain on each day data for other strains was acquired. After performing line scanning and alignment as described above, the GFP channel fluorescence intensity values between the SPB intensity peak locations in the mCherry channel, and plus end peak in the GFP channel, respectively, were compared using a Q-Q-plot. We computed the fluorescence $Fl$ (in arbitrary units) at quantiles $q$ from 0.5% to 99.5% in 0.5% increments from both datasets – the reference dataset we wanted to normalize to with fluorescence values $Fl_{\text{wt,ref}}(q)$, and the dataset we wanted to normalize from with fluorescence values $Fl_{\text{wt,from}}(q)$ – and fit the linear model $Fl_{\text{wt,ref}}(q) = \alpha \cdot Fl_{\text{wt,from}}(q) + \beta$ to estimate parameters $\alpha$ and $\beta$. The data from wildtype cells (wt) acquired on January 30th, 2018 (shown in *Figure 1b*) were set as the reference; wt (*Figure 1—figure supplement 1d*, left panel) and Kip2-S63A (*Figure 6f*) data from February 12th, 2018, as well as wt (*Figure 1—figure supplement 1d*, right panel), *bfa1Δ* and *bub2Δ* (*Figure 5—figure supplement 1*), and *bfa1Δbub2Δ* (*Figure 5d*) data from December 8th, 2018, were normalized using the corresponding wt data, respectively. The model fit yielded

$$\alpha = 0.6806 (95\% \text{CI} : 0.6796 - 0.6816)$$

$$\beta = 1957 (95\% \text{CI} : 1937 - 1978) \text{a.u.}$$

with $R^2 = 0.9997$ for the mapping from February 12th to January 30th data, and

$$\alpha = 0.9599 (95\% \text{CI} : 0.9563 - 0.9635),$$

$$\beta = 922.0 (95\% \text{CI} : 865.9 - 978.1) \text{a.u.},$$

with $R^2 = 0.9982$ for the mapping from December 8th to January 30th data, respectively.

For the purpose of demonstration and for calculating the speeds of fluorescent speckles, kymographs were generated and analyzed using Fiji. Shortly, a 5-pixel width line was placed along preanaphase cytoplasmic microtubules from SPBs towards plus-ends, and kymographs were created using the 'Reslice' function without interpolation. The position of the 5-pixel line was adjusted to cover the whole microtubule over time as well as possible. Due to the pivoting of cytoplasmic microtubules, these kymographs do not capture all fluorescent speckles from their origination to dissociation. When the microtubule moves out of the covered area, the corresponding speckles disappear from the kymograph. Conversely, when a part of the microtubule moves into the center of the covered area, dim speckles become brighter. The origination and dissociation of speckles were inspected in the time-lapse recordings frame by frame. The speed of fluorescent speckles moving towards microtubule plus-ends was calculated by extracting the coordinates of the starting and terminal positions of each speckle using the kymographs.

For fluorescence intensity, a Region Of Interest (ROI) was drawn around the area of interest (AOI) and the integrated density was extracted. An identically sized ROI was put next to the AOI to determine the background signal. The background intensity was subtracted from the ROI intensity to yield the fluorescence intensity (a.u.). For every experiment that was performed for quantification of

fluorescence intensity (a.u.), corresponding wild-type cells were imaged and analyzed for comparison to mutant cells. Average values of wild type cells of different experiments were used for normalization and comparison between experiments.

To determine the length of astral microtubules, endogenously expressed Bik1-3xGFP and Spc72-GFP were used as the plus- and minus-end marker, respectively. Three-dimensional coordinates of microtubule plus-ends and the corresponding SPBs were extracted with the Low Light Tracking Tool (*Krull et al., 2014*). The tracking tool does make mistakes when microtubules depolymerize with a very high rate, or when microtubules pivot quickly with large angles. Therefore, all tracked trajectories were inspected by eye to find and to correct those very rare mistakes. All of the time series tracking results were analyzed with custom functions written in Matlab (MathWorks). The distance between the b- and m- SPBs is the spindle length. Cells with spindles longer than 2 μm were excluded. The distance between a microtubule plus-end and the corresponding SPB represents the length of the microtubule. Only microtubules longer than 5-pixels (666.7 nm) were considered detectable due to the limit of the microscope resolution. Using this criterion, the maximum length and lifetime of each microtubule within the recorded time window (85.6 s) were extracted. Microtubule growth and shrinkage phases were annotated manually and recorded in Matlab, the speeds of microtubule growth and shrinkage were calculated using these annotations.

## Western blot

For protein extraction, 2 $OD_{600}$ log phase cell cultures were spun down and pellets were washed once with ice cold PBS, then lysed with Zirconia-Silicate beads in lysis buffer (50 mM Tris pH 7.5, 150 mM NaCl, 0.5 mM EDTA, 1 mM $MgCl_2$, Roche Complete Protease and phosphatase inhibitors and 0.2% NP- 40) on a FastPrep-24 homogenizer. Lysate was cleared by centrifugation at 5000 x *g*, 4°C for 5 min. Samples were separated on a 6% SDS-polyacrylamide gel (SDS-PAGE), wet-transferred onto a polyvinylidene fluoride (PVDF) membrane for western blotting. Antibodies used were primary antibodies anti-HA (1:1000, mouse monoclonal, Covance Inc), anti-GFP (1:1000, mouse monoclonal, Roche), anti-Pgk1 (1:4000, mouse monoclonal, Invitrogen), and secondary antibody goat anti-Mouse IgG conjugated to horseradish peroxidase (1:5000, Bio-Rad).

## Statistics

Each experiment was repeated with three or more independent clones (biological replicates). For wt strains, extra technical replicates were performed. The 95% confidence interval (95% CI) is shown in the graphs, or as indicated. n.s. (not significant) or asterisks indicate P values from Student's t-test or one-way ANOVA as indicated. Statistical analyses were performed on the means of the replicates.

## Acknowledgements

We thank J Hehl and the light microscopy center of ETH Zürich (ScopeM) for microscopy support; D Panozzo for assistance on image analysis; H-M Kaltenbach for input to the measurement model; A-M Farcas for assistance with western blotting. We acknowledge financial support by the SystemsX.ch RTD Grant #2012/192 TubeX (to MOS, JS and YB) and grants from the Swiss National Science Foundation (31003A-105904 to YB and 31003A_166608 to MOS).

## Additional information

### Funding

| Funder | Grant reference number | Author |
| --- | --- | --- |
| SystemsX.ch | RTD Grant #2012/192 TubeX | Michel O Steinmetz<br>Jörg Stelling<br>Yves Barral |
| Swiss National Science Foundation | 31003A-105904 | Yves Barral |
| Swiss National Science Foundation | 31003A_166608 | Michel O Steinmetz |

The funders had no role in study design, data collection and interpretation, or the decision to submit the work for publication.

## Author contributions

Xiuzhen Chen, Lukas A Widmer, Conceptualization, Data curation, Formal analysis, Validation, Investigation, Methodology, Writing—review and editing; Marcel M Stangier, Identified the critical motif for Kip2's ATPase activity; Michel O Steinmetz, Funding acquisition, Writing—review and editing, Identified the critical motif for Kip2's ATPase activity; Jörg Stelling, Yves Barral, Conceptualization, Supervision, Funding acquisition, Investigation, Methodology, Writing—original draft, Writing—review and editing

## Author ORCIDs

Xiuzhen Chen (iD) https://orcid.org/0000-0002-3027-6441
Lukas A Widmer (iD) https://orcid.org/0000-0003-1471-3493
Jörg Stelling (iD) https://orcid.org/0000-0002-1145-891X
Yves Barral (iD) https://orcid.org/0000-0002-0989-3373

## Decision letter and Author response

Decision letter https://doi.org/10.7554/eLife.48627.027
Author response https://doi.org/10.7554/eLife.48627.028

# Additional files

## Supplementary files

• Supplementary file 1. Source data for *Figure 4bcdefgi*, *Figure 5bceg*, *Figure 6dejk*, *Figure 7bd*, *Figure 1—figure supplement 1*, *Figure 2—figure supplement 2bd*, and *Figure 7—figure supplement 1b*.
DOI: https://doi.org/10.7554/eLife.48627.022

• Supplementary file 2. Strains used in this study.
DOI: https://doi.org/10.7554/eLife.48627.023

• Supplementary file 3. Stochastic simulation propensities and change vectors. Git code repository: https://gitlab.com/csb.ethz/Kip2-SPB-Profile-Manuscript (*Chen, 2019*; copy archived at https://github.com/elifesciences-publications/Kip2-SPB-Profile-Manuscript).
DOI: https://doi.org/10.7554/eLife.48627.024

## Data availability

All data and code are available in the main text, the supplementary materials, or at https://gitlab.com/csb.ethz/Kip2-SPB-Profile-Manuscript (copy archived at https://github.com/elifesciences-publications/Kip2-SPB-Profile-Manuscript).

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

## Appendix 1

DOI: https://doi.org/10.7554/eLife.48627.025

# Model: Mathematical Formulation

We formulate the following ordinary differential equations for the binding probability of kinesin motors at the minus-end $p_1$ , on the interior lattice $p_i$ , and at the plus-end $p_N$ of a 1D protofilament of length $N$:

$$\frac{dp_1}{dt} = (1-p_1)r_{\text{in}} + (1-p_1)r_{\text{on}} - p_1 k_{\text{off}} - p_1(1-p_2)k_{\text{step}} \tag{1}$$

$$\frac{dp_i}{dt} = p_{i-1}(1-p_i)k_{\text{step}} + (1-p_i)r_{\text{on}} - p_i k_{\text{off}} - p_i(1-p_{i+1})k_{\text{step}} \tag{2}$$

$$\frac{dp_N}{dt} = p_{N-1}(1-p_N)k_{\text{step}} + (1-p_N)r_{\text{on}} - p_N k_{\text{out}} \tag{3}$$

Here, $r_{\text{in}} = k_{\text{in}}[\text{Kip2}]_{\text{free}}$ and $r_{\text{on}} = k_{\text{on}}[\text{Kip2}]_{\text{free}}$, and the probabilities are subject to $0 \le p_1 \le 1,\, 0 \le p_i \le 1, 0 \le p_N \le 1$.

The microtubule filament was assumed to be static for analytical tractability. Note that while microtubule growth at in vivo speeds does not change the lattice increase and peak intensity of the predicted mean Kip2 distribution profile substantially, it does result in a wider Kip2 peak at the microtubule (*Figure 2—figure supplement 1*).

# Stochastic Simulation

To perform stochastic simulations, we use a direct-method Gillespie algorithm (*Gillespie, 1977*). Initially, all Kip2 is assumed to be free, that is $[\text{Kip2}]_{\text{free}} = [\text{Kip2}]_{\text{total}}$. The state $x$ of the protofilament is a binary vector of length $N$, with each entry describing a binding site $x_i$. Each binding site can either be free, $x_i = 0$, or occupied by a motor, $x_i = 1$, that is

$$\text{free}(x_i) := \begin{cases} 1 & x_i = 0 \\ 0 & \text{otherwise} \end{cases}, \text{and} \operatorname{occ}(x_i) := 1 - \text{free}(x_i)$$

The state then evolves according to the propensities and state change vectors given in *Supplementary file 3*.

# Model Parameters

### Lattice Loading Rate $r_{\text{on}}$

The lattice loading rate is given by $r_{\text{on}} = k_{\text{on}}[\text{Kip2}]_{\text{free}}$, where $k_{\text{on}}$ is the Kip2 motor on rate per binding site, per second, per nM Kip2. We start parameter search around the published in vitro on rate per microtubule, per minute, per nM Kip2 (*Roberts et al., 2014*). Since in vivo, both Bik1 and Bim1 are present, we sample around the value of 3.9 $\mu\text{m}^{-1}\text{min}^{-1}\text{nM}^{-1}$, which, assuming 13 protofilaments and 125 binding sites per micron of protofilament, that is 8 nm per binding site, translates to $k_{\text{on}} = 4 \times 10^{-5}\ \text{s}^{-1}\text{nM}^{-1}$. Samples were thus drawn from $1 \times 10^{-5}\ \text{s}^{-1}\text{nM}^{-1}$ up to 1 $\text{s}^{-1}\text{nM}^{-1}$ in 15 steps in log-space to cover a reasonable range of possible rate constants.

### Total Kip2 Concentration $[\text{Kip2}]_{\text{total}}$

Another unknown parameter is the Kip2 concentration $[\text{Kip2}]_{\text{total}}$, which we specify in nM units. Estimating this value is more involved – we start by querying PaxDB (*Wang et al., 2015*), a database that aggregates protein abundances for different organisms in parts per million, for Kip2 abundance. From the experimental estimates in PaxDB (discarding the integrated

model), we compute the median Kip2 abundance in ppm, 14.6 ppm. This number can then be converted to proteins per micron cubed by multiplying by $2.5 \times 10^6$ proteins $\mu m^{-3}$ as derived for *S. cerevisiae* (*Milo, 2013*), to yield an estimated 36.5 Kip2 molecules per micron cubed. The volume of a yeast cell in preanaphase is approximately 50 $\mu m^3$ (*Uchida et al., 2011*). Thus, approximately 1825 Kip2 molecules are available in a preanaphase yeast cell, corresponding to a concentration of 61 nM. If we consider a single microtubule of 13 protofilaments, there are about 140 Kip2 molecules available per protofilament. We use this number as a rough guess for parameter search in our single-protofilament model, and draw samples uniformly from 60 to 200 Kip2 molecules per protofilament in 20 molecule increments, and then 300, 400 and 500 molecules per protofilament to investigate the effect of higher concentrations of up to 218 nM.

## Minus-End Loading Rate $r_{in}$

The minus-end loading rate was unknown at first. However, it is useful to compare the minus-end loading rate $r_{in} = k_{in}[\text{Kip2}]_{\text{free}}$ with the overall loading rate on the lattice

$$r_{\text{on,total}} = \sum_{i=1}^{N} r_{\text{on}} = \sum_{i=1}^{N} k_{\text{on}}[\text{Kip2}]_{\text{free}} = N k_{\text{on}}[\text{Kip2}]_{\text{free}}$$

where $r_{in} \gg r_{\text{on,total}}$ if and only if $k_{in} \gg N k_{\text{on}}$. Since $k_{\text{off}} \approx 0$, no motors can detach on the way, and we can distinguish between minus-end-dominated and lattice-dominated motor binding by comparing the in- and on-rates. If $k_{in}/k_{\text{on}} \gg N$, then motor binding is minus-end-dominated, if $k_{in}/k_{\text{on}} \ll N$, then motor binding is lattice-dominated. For initializing the parameter search, we can thus explore the regime around $k_{in} \approx N k_{\text{on}}$. For a 2nm microtubule, $N = 250$, that is for the in rate constant to have a measurable effect, it should roughly be two orderorders of magnitudes higher than the on rate constant. Therefore, the in rate constant was sampled in 15 steps logarithmically in a range shifted up by two orders of magnitude, that is from $1 \times 10^{-3}$ $s^{-1}nM^{-1}$ to $1 \times 10^2$ $s^{-1}nM^{-1}$.

## Stepping rate $k_{\text{step}}$

The stepping rate $k_{\text{step}}$ can be inferred from the velocity the motor shows while not encountering any obstacle, if we assume 8 nm steps (corresponding to the length of a tubulin heterodimer). The free-stepping motor velocity in vitro was quantified in the presence of the proteins Bim1 and Bik1, Kip2 binding partners in vivo (*Roberts et al., 2014*). Since measurements of speckle speeds from our own kymographs were available, we fixed the stepping rate in the model to the mean measured value of $k_{\text{step}} = 13.04$ $s^{-1}$.

## Lattice off rate $k_{\text{off}}$

The lattice off rate $k_{\text{off}}$ was previously determined in vitro (*Roberts et al., 2014*), both in the absence of Bim1 and Bik1 (0.473 $s^{-1}$), as well as their presence ($7.3 \times 10^{-3}$ $s^{-1}$). However, these experiments were performed without crowding agents, and kinesin motors tend to be more processive in crowded environments (*Conway and Ross, 2014*). The microtubules we consider are shorter than 4 $\mu m$, but, on average, a motor is expected to run around 8 $\mu m$ without falling off, and potentially even further in the crowded cytoplasm in vivo. We therefore fix the off rate in the model to 0 $s^{-1}$.

## Plus-end off rate $k_{\text{out}}$

The plus-end off rate $k_{\text{out}}$ is conceptually different from the lattice off rate $k_{\text{off}}$ and the stepping rate $k_{\text{step}}$ because the motor cannot simply step off the end, and it does not stay bound forever. An off rate of $2.27 \times 10^{-2}$ $s^{-1}$ for single motors has previously been determined in vitro (*Hibbel et al., 2015*). However, it is unclear if multiple motors at the plus-tip would detach

more readily. In the limiting case of the plus-end off rate being equal to the step rate, the motors would simply step off past the microtubule with the same speed as they move on the lattice. Therefore, we investigated the parameter range between the single-motor off rate, and the stepping rate: we sampled $k_{\text{out}}$ linearly from 1 s$^{-1}$ up to the value of $k_{\text{step}}$ of 13.04 s$^{-1}$ in 1.0033 s$^{-1}$ increments.

## Conditions for Flat Kinesin Profiles

To derive a condition on the parameters required for a flat mean motor occupancy profile, we require that the probability of a motor being bound on the lattice is equal for all lattice sites in the midzone, that is the zone between the microtubule minus- and plus-ends. The equations for such a system read as follows:

$$\frac{dp_1}{dt} = (1 - p_1)r_{\text{in}} + (1 - p_1)r_{\text{on}} - p_1 k_{\text{off}} - p_1(1 - p_M)k_{\text{step}} \tag{4}$$

$$\frac{dp_M}{dt} = p_1(1 - p_M)k_{\text{step}} + (1 - p_M)r_{\text{on}} - p_M k_{\text{off}} - p_M(1 - p_M)k_{\text{step}} \tag{5}$$

$$\frac{dp_M}{dt} = p_M(1 - p_M)k_{\text{step}} + (1 - p_M)r_{\text{on}} - p_M k_{\text{off}} - p_M(1 - p_M)k_{\text{step}} \tag{6}$$

$$\frac{dp_M}{dt} = p_M(1 - p_M)k_{\text{step}} + (1 - p_M)r_{\text{on}} - p_M k_{\text{off}} - p_M(1 - p_N)k_{\text{step}} \tag{7}$$

$$\frac{dp_N}{dt} = p_M(1 - p_N)k_{\text{step}} + (1 - p_N)r_{\text{on}} - p_N k_{\text{out}} \tag{8}$$

where the probabilities are subject to $0 \leq p_1 \leq 1$, $0 \leq p_M \leq 1$, $0 \leq p_N \leq 1$. We are interested in the steady-state profile generated. Therefore, we set the left-hand side of above equation system to zero and derive the steady-states $p_i^\star$, depending on the parameters. $p_i^{(0)}$ refers to the initial state of $p_i$.

Case 1: $k_{\text{step}} = 0$

If $k_{\text{step}} = 0$, it follows immediately that all the equations become uncoupled, that is each lattice binding site becomes an independent binding site. For the minus-end site, unless $r_{\text{in}} = r_{\text{on}} = k_{\text{off}} = 0$ and $p_1^\star = p_1^{(0)}$ (there are no dynamics),

$$p_1^\star = \frac{r_{\text{in}} + r_{\text{on}}}{r_{\text{in}} + r_{\text{on}} + k_{\text{off}}}$$

For the inner lattice sites, unless $r_{\text{on}} = k_{\text{off}} = 0$ and $p_M^\star = p_M^{(0)}$,

$$p_M^\star = \frac{r_{\text{on}}}{r_{\text{on}} + k_{\text{off}}}$$

For the plus-end site, unless $r_{\text{on}} = k_{\text{out}} = 0$ and $p_N^\star = p_N^{(0)}$,

$$p_N^\star = \frac{r_{\text{on}}}{r_{\text{on}} + k_{\text{out}}}$$

Case 2: $k_{\text{step}} > 0, k_{\text{out}} = 0$

This scenario consists of a permanent roadblock at the plus end. From

$$p_M^\star(1 - p_N^\star)k_{\text{step}} + (1 - p_N^\star)r_{\text{on}} - \underbrace{p_N^\star k_{\text{out}}}_{=0} = 0$$

it follows that $(1 - p_N^\star)(r_{\text{on}} + p_M^\star k_{\text{step}}) = 0$. This implies

- $p_N^\star = 1$, and/or

- $r_{\mathrm{on}} = 0$ and $p_M^* = 0$. $p_N^*$ is then the same as the initial condition $p_N^{(0)}$. From

$$\underbrace{p_1^*(1-p_M^*)k_{step}}_{=p_1^* k_{step}} + \underbrace{(1-p_M^*)r_{on}}_{=0} - \underbrace{(1-p_M^*)k_{off}}_{=0} - \underbrace{p_M^*(1-p_M^*)k_{step}}_{=0} = 0$$

it follows that $p_1^* = 0$. Using

$$\underbrace{(1-p_1^*)r_{in}}_{=r_{in}} + \underbrace{(1-p_1^*)r_{out}}_{=0} - \underbrace{p_1^* k_{off}}_{=0} - \underbrace{p_1^*(1-p_M^*)k_{step}}_{=0} = 0$$

immediately shows that $r_{\mathrm{in}} = 0$.

If $p_N^* = 1$, the plus-end site is fully occupied. Then equations

$$p_M^*\left(1-p_M^*\right)k_{step} + \left(1-p_M^*\right)r_{on} - p_M^* k_{off} - p_M^* - \underbrace{p_M^*\left(1-p_N^*\right)}_{=0}k_{step} = 0 \text{ and}$$

$$p_M^*\left(1-p_M^*\right)k_{step} + \left(1-p_M^*\right)r_{on} - p_M^* k_{off} - p_M^*\left(1-p_M^*\right)k_{step} = 0$$

imply that either of two cases must hold:

1. $p_M^* = 0$ and $r_{\mathrm{on}} = 0$. It then again follows that $p_1^*\left(1-p_M^*\right)k_{step} = 0$ and thus $p_1^* = 0$, which in turn implies that $r_{\mathrm{in}} = 0$.
2. $p_M^* = 1$ and $k_{\mathrm{off}} = 0$. In this case, the motors arrive from the minus-end and get stuck on the lattice, never reaching the plus end. Then,

$$\left(1-p_1^*\right)r_{\mathrm{in}} + \left(1-p_1^*\right)r_{\mathrm{on}} - \underbrace{p_1^* k_{off}}_{=0} - \underbrace{p_1^*(1-p_M^*)k_{step}}_{=0} = 0$$

either implies $p_1^* = 1$, or $r_{\mathrm{in}} + r_{\mathrm{on}} = 0$, in which case $p_1^* = p_1^{(0)}$.

Case 3: $k_{\mathrm{step}} > 0, k_{\mathrm{out}} > 0, k_{\mathrm{in}} = 0, k_{\mathrm{on}} = 0$

In this case, the steady-state equation for $p_1^*$ reads

$$\underbrace{(1-p_1^*)r_{in}}_{=0} + \underbrace{(1-p_1^*)r_{on}}_{=0} - p_1^* k_{off} - p_1^*\left(1-p_M^*\right)k_{step} = 0 \Leftrightarrow p_1^*\left(k_{off} + \left(1-p_M^*\right)k_{step}\right) = 0$$

Two sub-cases fulfill this condition:

1. $p_M^* = 1$ and $k_{\mathrm{off}} = 0$, in which case $p_1^* = p_1^{(0)}$. This corresponds to case two above, where $r_{\mathrm{in}} + r_{\mathrm{on}} = 0$, and then

$$\underbrace{p_M^*(1-p_M^*)k_{Step}}_{=0} + \underbrace{(1-p_M^*)r_{on}}_{=0} - \underbrace{p_M^* k_{off}}_{=0} - \underbrace{p_M^*(1-p_M^*)k_{step}}_{=(1-p_N^*)k_{Step}} = 0$$

implies that $p_N^* = 1$. Thus, the lattice is fully occupied, and any motor bound at the minus-end site cannot move.

2. $p_1^* = 0$, that is the first binding site is always empty. Then,

$$\underbrace{p_1^*(1-p_M^*)k_{step}}_{=0} + \underbrace{(1-p_M^*)r_{on}}_{=0} - p_M^* k_{off} - p_M^*\left(1-p_M^*\right)k_{step} = 0$$

$$\Leftrightarrow p_M^*\left(k_{off} + \left(1-p_M^*\right)k_{step}\right) = 0$$

Then, there are again two cases:

a. $p_M^* = 0$. Then,

$$\underbrace{p_M^*(1-p_N^*)k_{step}}_{=0} + \underbrace{(1-p_N^*)r_{on}}_{=0} - p_N^* k_{out} = 0,$$

which implies $p_N^* = 0$, that is all the sites are empty.

b. $p_M^* = 1$ and $k_{\mathrm{off}} = 0$. Then

$$\underbrace{p_M^*(1-p_M^*)k_{step}}_{=0} + \underbrace{(1-p_M^*)r_{on}}_{=0} - \underbrace{p_M^*k_{off}}_{=0} - \underbrace{p_M^*(1-p_N^*)k_{step}}_{=(1-p_N^*)k_{step}} = 0$$

implies that $p_N^* = 1$, that is the first site is empty, and the lattice and plus-end sites are occupied.

Case 4: $k_{step}>0$, $k_{out}>0$, $k_{in} + k_{on}>0$

From the steady-state equations

$$
\begin{aligned}
\left(1-p_1^*\right)r_{in} + \left(1-p_1^*\right)r_{on} - p_1^*k_{off} - p_1^*\left(1-p_M^*\right)k_{step} &= 0\\
\left(1-p_1^*\right)(r_{in} + r_{on}) - p_1^*\left(k_{off} + \left(1-p_M^*\right)k_{step}\right) &= 0\\
p_M^*(1-p_M^*)k_{step} + (1-p_M^*)r_{on} - p_M^*(1-p_M^*)k_{step} &= 0\\
\Leftrightarrow -k_{step}p_M^{*2} + k_{step}p_M^* + r_{on} - p_M^*r_{on} - p_M^*k_{step} + p_M^{*2}k_{step} &= 0\\
\Leftrightarrow r_{on} - p_M^*r_{on} - p_M^*k_{off} &= 0
\end{aligned}
$$

If $r_{on} + k_{off}>0$, then

$$p_M^* = \frac{r_{on}}{r_{on} + k_{off}}$$

For the plus-end site, this leads to:

$$
\begin{aligned}
p_M^*\left(1-p_M^*\right)k_{step} + \left(1-p_M^*\right)r_{on} - p_M^*k_{off} - p_M^*\left(1-p_N^*\right)k_{step} &= 0\\
\Leftrightarrow p_N^* = p_M^* + \frac{k_{off}+r_{on}}{k_{step}} - \frac{r_{on}}{k_{step}p_M^*}\\
\Leftrightarrow p_N^* = \frac{r_{on}}{r_{on}+k_{off}} + \frac{k_{off}+r_{on}}{k_{step}} - \frac{r_{on}}{k_{step}}\frac{r_{on}+k_{off}}{r_{on}}\\
\Leftrightarrow p_N^* = \frac{r_{on}}{r_{on}+k_{off}}
\end{aligned}
$$

From

$$
\begin{aligned}
p_M^*\left(1-p_N^*\right)k_{step} + \left(1-p_N^*\right)r_{on} - p_N^*k_{out} &= 0\\
\frac{r_{on}}{r_{on}+k_{off}}\frac{k_{off}}{r_{on}+k_{off}}k_{step} + \frac{k_{off}}{r_{on}+k_{off}}r_{on} - \frac{r_{on}}{r_{on}+k_{off}}k_{out} &= 0
\end{aligned}
$$

it follows that

$$k_{out} = k_{off}\frac{r_{on} + k_{off} + k_{step}}{r_{on} + k_{off}}.$$

Then, for the minus end site,

$$
\begin{aligned}
(1-p_1^*r_{in}) + (1-p_1^*)r_{on} - 1 - p_1^*k_{off} - 1 - p_1^*(1-1-p_M^*k_{step}) &= 0\\
\frac{k_{off}}{r_{on}+k_{off}}r_{in} + \frac{k_{off}}{r_{on}+k_{off}}r_{on} - \frac{k_{on}}{r_{on}+k_{off}}r_{off} - \frac{k_{on}}{r_{on}+k_{off}}\frac{k_{off}}{r_{on}+k_{off}}k_{step} &= 0\\
k_{off}r_{in} + k_{off}r_{on} - r_{on}k_{off} - \frac{r_{on}k_{off}k_{step}}{r_{on}+k_{off}} &= 0\\
\Leftrightarrow r_{in} = \frac{r_{on}k_{step}}{r_{on}+k_{off}}
\end{aligned}
$$

This holds in two cases:

- $k_{off} = 0$, $r_{on}>0$, which implies $k_{out} = 0$ and thus is covered in case 2, and/or
- $p_1^* = \frac{r_{on}}{r_{on}+k_{off}}$. Then, from

$$
\begin{aligned}
\left(1-p_1^*\right)r_{in} + \left(1-p_1^*\right)r_{on} - p_1^*k_{off} - p_1^*\left(1-p_M^*\right)k_{step} &= 0\\
\frac{k_{off}}{r_{on}+k_{off}}r_{in} + \frac{k_{off}}{r_{on}+k_{off}}r_{on} - \frac{r_{on}}{r_{on}+k_{off}}k_{off} - \frac{r_{on}}{r_{on}+k_{off}}\frac{k_{off}}{r_{on}+k_{off}}k_{step} &= 0\\
k_{off}r_{in} + k_{off}r_{on} - r_{on}k_{off} - \frac{r_{on}k_{off}k_{step}}{r_{on}+k_{off}} &= 0\\
\Leftrightarrow r_{in} = \frac{r_{on}k_{step}}{r_{on}+k_{off}}
\end{aligned}
$$

Thus, for $r_{on} + k_{off}>0$, we have

$$p^* = p_1^* = p_M^* = p_N^* = \frac{r_{on}}{r_{on} + k_{off}}.$$

The flow rate of motors from one site to the next is accordingly

$$p^*(1-p^*)k_{\text{step}} = \frac{r_{\text{on}}k_{\text{off}}}{(r_{\text{on}}+k_{\text{off}})^2}k_{\text{step}}.$$

This is the same as the inflow at the minus-end

$$
\begin{aligned}
(1-p^*)(r_{\text{in}}+r_{\text{on}}) - p^*k_{\text{off}} &= \frac{k_{\text{off}}}{r_{\text{on}}+k_{\text{off}}}(r_{\text{in}}+r_{\text{on}}) - \frac{r_{\text{on}}}{r_{\text{on}}+k_{\text{off}}}k_{\text{off}} \\
&= \frac{\frac{r_{\text{on}}k_{\text{step}}}{r_{\text{on}}+k_{\text{off}}}k_{\text{off}}}{r_{\text{on}}+k_{\text{off}}} \\
&= \frac{r_{\text{on}}k_{\text{off}}}{(r_{\text{on}}+k_{\text{off}})^2}k_{\text{step}}
\end{aligned}
$$

In contrast, the outflow at the plus-end is

$$
\begin{aligned}
p^*k_{\text{out}} &= \frac{r_{\text{on}}}{r_{\text{on}}+k_{\text{off}}}k_{\text{off}}\frac{r_{\text{on}}+k_{\text{off}}+k_{\text{step}}}{r_{\text{on}}+k_{\text{off}}} \\
&= \frac{r_{\text{on}}k_{\text{off}}}{(r_{\text{on}}+k_{\text{off}})^2}(r_{\text{on}}+k_{\text{off}}+k_{\text{step}}) \\
&= p^*(1-p^*)k_{\text{step}} + (1-p^*)r_{\text{on}}
\end{aligned}
$$

If $r_{\text{on}}+k_{\text{off}}=0$, it follows that $r_{\text{in}}>0$, and

$$
\begin{aligned}
0 &= (1-p_1^*)r_{\text{in}} - p_1^*(1-p_M^*)k_{\text{step}} \\
0 &= (p_1^*-p_M^*)(1-p_M^*)k_{\text{step}} \\
0 &= p_M^*(p_N^*-p_M^*)k_{\text{step}} \\
0 &= p_M^*(1-p_N^*)k_{\text{step}} - p_N^*k_{\text{out}}
\end{aligned}
$$

There are again multiple solutions to this system of equations:

- $p_1^* = p_M^* = p_N^* = 1$, where $k_{\text{out}}=0$ and $r_{\text{in}}$ is arbitrary (but >0 per assumption).
- $p_1^* = p_M^* = p_N^* = p^* < 1$. Then, we get $p^* = \frac{r_{\text{in}}}{k_{\text{step}}}$, $r_{\text{in}}<k_{\text{step}}$ and $k_{\text{out}} = k_{\text{step}} - k_{\text{in}}$. The flow rate of motors from one site to the next then is $p^*(1-p^*)k_{\text{step}} = \frac{r_{\text{in}}(k_{\text{step}}-r_{\text{in}})}{k_{\text{step}}}$. The inflow at the minus end is $(1-p^*)r_{\text{in}} = \frac{r_{\text{in}}(k_{\text{step}}-r_{\text{in}})}{k_{\text{step}}}$, and the outflow at the plus end is $p_N^*k_{\text{out}} = \frac{r_{\text{in}}(k_{\text{step}}-r_{\text{in}})}{k_{\text{step}}}$. This is the solution that is most consistent with the parameter values and observed behavior for the Kip2-3xsfGFP mean profile in wildtype cells.

## Measurement Model

The measurement model maps fluorescent Kip2 molecules on the protofilament lattice of the in silico model to in vivo fluorescence signals as measured by a confocal microscope. Since the underlying microtubule length $l$ resulting in the minus-end-peak to plus-end-peak length $l_{\text{peak}}$ is a priori unknown, we simulate a population of microtubules with lengths ranging from 1.112 µm to 3.136 µm in 88 nm increments. This was deemed sufficiently accurate to cover the measured bin lengths (that range from 1.333 µm to 2.933 µm) by inspecting the fraction of profiles contributing to each bin, and by sampling more densely. The simulation and measurement model computation then proceed as follows:

1. Stochastic simulation
    Simulate model as detailed in the stochastic simulation section for two hours of simulation time.
2. Sampling of time points from simulation at 0.25 Hz (1 Hz for **Figure 2B**) for one hour of simulation time after initial burn-in of one hour simulation time.
3. Convolution of sampled profiles with 1D point-spread function (PSF). We use a 1D Gaussian PSF with

$$\sigma = 0.21\frac{\lambda}{NA},$$

where $\lambda \approx 509$ nm is the emission wavelength of the used GFP fluorophore, and $NA$ = 1.46 is the numerical aperture of the objective lens used (**Thomann et al., 2002**).
4. Pixel sampling and binning
    Each pixel has a side length of 133 nm, which corresponds to approximately 17 motor

binding sites of 8 nm on the protofilament. We compute all mappings from binding sites to pixels, with integer offsets of binding sites, by averaging the fluorescence values of the binding sites in each pixel, resulting in 17 possible samplings of each convolved profile.

5. Peak detection

For detecting the location of the microtubule plus-end, as for the in vivo data, the microtubule plus-end was defined by finding the local fluorescence maximum at the microtubule plus-tip.

6. Restriction to profile lengths within bin

Since the in vivo profiles were analyzed by binning their lengths from the minus-end to the plus-end with a ± 1 pixel threshold, model profiles were binned analogously.

7. Centered within-bin alignment and mean computation

Finally, all the profiles within a bin are aligned by centering them, and the bin mean is computed.

Step 7 generates the final model mean $\mu_{\mathrm{model}}(x_{i,\mathrm{model}}, \mathbf{p})$ with sample locations $x_{i,\mathrm{model}}$ and parameters $\mathbf{p}$.

## Likelihood Function and Confidence Regions

We compare the mean profile $\mu_{\mathrm{data}}(x_{i,\mathrm{data}})$ with standard error $\mathrm{SEM}(x_{i,\mathrm{data}})$ at locations $x_{i,\mathrm{data}}$ obtained from in vivo data, with the mean profile

$$\tilde{\mu}_{\mathrm{model}}(x_{i,\mathrm{model}}, \mathbf{p}, A, B) = A + B \cdot \mu_{\mathrm{model}}(x_{i,\mathrm{model}}, \mathbf{p})$$

with parameters $\mathbf{p}$ at locations $x_{i,\mathrm{model}}$, by linearly interpolating $\tilde{\mu}_{\mathrm{model}}(x_{i,\mathrm{model}}, \mathbf{p}, A, B)$ at $x_{i,\mathrm{data}}$ and thus estimating $\tilde{\mu}_{\mathrm{model}}(x_{i,\mathrm{data}}, \mathbf{p}, A, B)$. $A$ is given by the mean background fluorescence, and $B$ is the maximum mean fluorescence corresponding to a fully occupied lattice, which can be estimated from the saturation of the (normalized) Kip2-S63A mutant fluorescence (at approximately 13.7 a.u. when optimized over all experimental conditions fit with the model). We assume a normal error for the data, which gives rise to the likelihood function

$$L(\mathrm{model}|\mathrm{data}) = \frac{1}{(2\pi)^{N/2}} \prod_{i=1}^{N} \mathrm{SEM}(x_{\mathrm{data},i})^{-1} \cdot \prod_{i=1}^{N} e^{-\frac{\left(\tilde{\mu}_{\mathrm{model}}(x_{\mathrm{data},i}, \mathbf{p}, A, B) - \mu_{\mathrm{data}}(x_{\mathrm{data},i})\right)^2}{2\mathrm{SEM}(x_{\mathrm{data},i})^2}}$$

In order to find the best-fitting parameter set, rather than maximizing the likelihood directly, we minimize the negative log-likelihood:

$$\begin{aligned} -2LL \ &= -2\left(-\tfrac{N}{2}\log(2\pi) - \sum_{i=1}^{N}\mathrm{SEM}(x_{\mathrm{data},i}) - \sum_{i=1}^{N}\frac{\left(\tilde{\mu}_{\mathrm{model}}(x_{\mathrm{data},i}, \mathbf{p}, A, B) - \mu_{\mathrm{data}}(x_{\mathrm{data},i})\right)^2}{2\mathrm{SEM}(x_{\mathrm{data},i})^2}\right) \\ &= \underbrace{N\log(2\pi) + 2\sum_{i=1}^{N}\mathrm{SEM}(x_{\mathrm{data},i})}_{=const.=:C} + \underbrace{\sum_{i=1}^{N}\frac{\left(\tilde{\mu}_{\mathrm{model}}(x_{\mathrm{data},i}, \mathbf{p}, A, B) - \mu_{\mathrm{data}}(x_{\mathrm{data},i})\right)^2}{\mathrm{SEM}(x_{\mathrm{data},i})^2}}_{=:SSE(\mathbf{p})} \end{aligned}$$

Since $C$ is a constant, we can simply minimize the sum-of-squares term over all possible parameter sets $\mathbf{p}$ in order to find the maximum likelihood parameter set $\mathbf{p}^{\mathrm{ML}}$:

$$\mathbf{p}^{\mathrm{ML}} := \underset{\mathbf{p}}{arg\,min}(SSE(\mathbf{p}))$$

Finally, we sample from the posterior by first sampling which bin a microtubule belongs to proportionally to the number of microtubule profiles in each bin, and then proportionally to the likelihood $e^{-\frac{1}{2}SSE(\mathbf{p})}$.

