## [Decision Letter]

Thank you for submitting your article "Remote control of microtubule plus-end dynamics and function from the minus-end" for consideration by *eLife*. Your article has been reviewed by three peer reviewers, one of whom is a member of our Board of Reviewing Editors, and the evaluation has been overseen by Vivek Malhotra as the Senior Editor. The reviewers have opted to remain anonymous.

The reviewers have discussed the reviews with one another and the Reviewing Editor has drafted this decision to help you prepare a revised submission.

Summary:

In eukaryotic cells, intracellular components are positioned in space and time in part due to the microtubule cytoskeleton. Cellular microtubules are functionally diverse, despite being formed from a common pool of tubulin dimers. In asymmetrically dividing cells the microtubules associated with the old and new centrosome are often functionally distinct. However, mechanistically what leads to these distinctions is not well understood in any system. In the current work, Chen and Widmer et al. use a combination of in vivo imaging in *S. cerevisiae* and mathematical modelling to determine that the differing cargoes and plus end dynamics of bud-microtubules and mother-microtubules is specified at their minus ends (found anchored in the spindle pole body (SPB)). They find that old SPBs recruit the kinesin Kip2 to the minus end in a manner that depends on Bub2 and Bfa1 as well as the phosphorylation state of Kip2. Kip2 then translocates to the plus end, transporting dynein to the bud and promoting microtubule extension. The authors conclude this rigorously performed study by proposing a model for how microtubule organizing centers could differentially specify the plus end behavior of the microtubules emanating from them.

Essential revisions:

1) One of the authors' main conclusions from their modeling is that Kip2 starts its runs from the b-SPB and ends them at the microtubule plus end. However, this is really difficult to visualize in the experimental kymograph shown in Figure 3A (right) and Figure 2—figure supplement 1A (middle). Specifically, in Figure 3A, some runs appear to be truncated halfway (e.g. 5th arrowhead from the left in the experimental kymograph). Some speckles appear to start in the microtubule lattice, not from SPB, although it is difficult to interpret from the blurry kymograph. Additionally, the synthetic kymograph does not closely resemble the experimental kymograph, as stated by the authors in the first paragraph of the subsection “Kip2 runs start from SPBs”. Thus, the claim "the initiation of Kip2 runs is restricted to the minus-ends" is weak. Clearer kymographs or additional experimental evidence is necessary to support this claim. Intensity line scans at different time points could help to more clearly distinguish individual motor runs in space and time.

2) Related to the first point, additional information should also be added to the Materials and methods describing how runs were determined. If multiple people analyzed this data, did they come to the same conclusions about what to consider a run? Was this data blinded in some way to account for potential bias in the analysis? Automated detection of runs would ease concerns that runs are being mis-assigned.

3) Mutating G374 in Kip2 is a nice second independent test for whether Kip2 starts its runs from the microtubule minus-ends (subsection “Kip2 runs start from SPBs”, first paragraph). However, if Kip2 is indeed different from Kip3 with regard to where it initiates runs, then mutating the corresponding residue in Kip3 (G343 based on the alignment in Figure 3—figure supplement 1) would be a good control to perform. The mutated Kip3-G343 would be predicted to decorate along the microtubule length and not accumulate at the b-SPB like Kip2. This would strengthen the argument that loading at SPBs and initiating runs at the minus end is a Kip2-specific phenomenon.

4) In Figure 5B, the authors show that *bfa1* and *bub2* deletions lowered the levels of Kip2-G374A on the b-SPB by nearly 50%, while the signal on the m-SPB was unchanged. Similarly, in Figure 5F, the levels of Kip2 on b-microtubules lowered to nearly m-microtubules levels, while the signals on the m-microtubules remained unchanged compared to wild type. These decreases might indicate that there is actually less total Kip2 in *bfa1* and *bub2* null cells. The authors should determine the expression levels of Kip2 by western blot to exclude the possibility that the decreases are not due to a reduction in Kip2's expression or protein stability.

5) In a related point, following the analysis of Kip2-S63A-G374A in Figure 6C, D, E, the authors conclude that both SPBs recruited more of the hypo-phosphorylated, ATPase deficient protein and that the distribution was still asymmetric in cells with correctly oriented SPBs. However, based on Figure 6D, it seems that the enhancement in both SPBs indicates that the total amount of Kip2 in the cell might have increased significantly because of the S63A mutation. The authors should compare the levels of Kip2-S63A-G374A to Kip2-G374A to exclude the possibility that dephosphorylation of Kip2's N-terminus enhances the stability of the protein, thereby giving the observed effects.

6) The authors' main conclusions concern Kip2's distribution profile in metaphase. However, they selected spindles for analysis and decided which spindles were in metaphase based on the shape of cells and the size of spindles (subsection “Image and data analysis”, first paragraph). The standard in the field for examining cells in metaphase is to perform cdc20 depletion (such as Khmelinskii et al., 2009). Given that metaphase is not strictly defined by the length of the spindle, it would be more appropriate to refer to preanaphase instead of metaphase.

7) Kip2 hypo-phosphorylation caused an increase in the levels of Kip2 on the plus ends of both b- and m-microtubules compared to wild type, based on Figure 6J versus Figure 4E, but the levels of dynein to the b-microtubules, a cargo of Kip2, did not appear to increase based on the images shown in Figure 7C (comparing the dynein dots in *KIP2* vs. *KIP2-S63A* images). On the other hand, in Figure 7D, the authors report that dynein distribution at the tip of b- and m-microtubules was randomized when Kip2 phosphorylation is prevented (i.e. in *KIP2-S63A* mutant cells). Please clarify how randomization occurred when there is no apparent change in dynein intensity?

8) Related to the modeling.

- The presented model addresses binding, unbinding and stepping rates of the kinesin. However, it does not consider dynamic microtubules (e.g. compare left vs. right kymographs in Figure 3A, where the plus end clearly grows in the experimental condition, but not in the model). Addition of plus end dynamics and showing that the regulation of Kip2 loading at the minus ends affects it would make the model predictions much stronger. For instance, how would the model predict observations presented in Figure 4?

- The total intensity at the tip corresponds to a defined accumulation of motors at the end. How many are there? (The number of occupied sites should be set by the ratio of minus-end loading and plus-end off rate; and can thus also be directly predicted from the model parameters.)

9) Points to discuss:

- Is it possible to exclude direct end-targeting of Kip2 (e.g. through Bim1 binding or another mechanism)? In principle, such a model would also produce a non-microtubule-length dependent Kip2 profile, here the *r_on_* would be the on rate of direct binding to the tip.

- If enhanced loading to the SPB is a specific mechanism for the b-SPB and not the m-SPB, the prediction would be that the distribution of Kip2 intensities is length-dependent on m-SPB-emanating microtubules. Is that true?

---

## [Author Response]

Essential revisions:1) One of the authors' main conclusions from their modeling is that Kip2 starts its runs from the b-SPB and ends them at the microtubule plus end. However, this is really difficult to visualize in the experimental kymograph shown in Figure 3A (right) and Figure 2—figure supplement 1A (middle). Specifically, in Figure 3A, some runs appear to be truncated halfway (e.g. 5th arrowhead from the left in the experimental kymograph). Some speckles appear to start in the microtubule lattice, not from SPB, although it is difficult to interpret from the blurry kymograph. Additionally, the synthetic kymograph does not closely resemble the experimental kymograph, as stated by the authors in the first paragraph of the subsection “Kip2 runs start from SPBs”. Thus, the claim "the initiation of Kip2 runs is restricted to the minus-ends" is weak. Clearer kymographs or additional experimental evidence is necessary to support this claim. Intensity line scans at different time points could help to more clearly distinguish individual motor runs in space and time.

The point of the reviewers is well taken. Indeed, kymographs are noisy and are therefore not easy to interpret. This may give the feeling that some traces start or stop abruptly. However, we provide the videos on which these kymographs are based. They help well interpreting the kymographs and see that the trains are indeed continuous, even if the pointed track seems jumpy. Altogether, the overall appearance of the Kip2 kymographs remains quite distinctive, particularly when comparing typical Kip2-3sfGFP kymographs (Figure 3A and Figure 2—figure supplement 1A) to those obtained with Kip2-S63A-3sfGFP and Kip3-3sfGFP (Figure 2—figure supplement 1A). Whereas in the last ones the signal is very low near the SPB and increases progressively towards the microtubule plus end (not considering the strictly plus-end signal itself), in all Kip2-3sfGP kymographs the signal is overall fairly uniform along microtubules. Thus, we find them still quite telling. We have clarified what can be obtained from them in the text.

Due to technical limitations of in vivo imaging (movement of the microtubule in space, signal to noise ratio…), we are not in a position yet to produce much clearer kymographs at this time. However, we thank the reviewers for their suggestion of using line scans. It indeed helps a lot visualizing the process. We have now added the frames of a video and the corresponding line scan, which nicely illustrate the motion of a Kip2 train from the minus to the plus end of a microtubule (Figure 3B). We also provide representative video recordings of Kip2-3xsfGFP, with which one can follow the moving speckles regardless of the pivoting of microtubules. In these videos, one can clearly see that the speckles start from SPBs and end at microtubule plus ends.

We agree with the reviewer that the synthetic kymograph is not very demonstrative and might be misleading since the model is computed on a static microtubule. Therefore, we have removed it from the figure.

2) Related to the first point, additional information should also be added to the Materials and methods describing how runs were determined. If multiple people analyzed this data, did they come to the same conclusions about what to consider a run? Was this data blinded in some way to account for potential bias in the analysis? Automated detection of runs would ease concerns that runs are being mis-assigned.

We determined the runs manually by closely following moving speckles with the video recordings. When a speckle was observed, we followed the speckle in the video backward frame by frame to determine where it originated. This analysis was systematically performed by Xiuzhen Chen according to the following scenario: While she evaluated all usable videos, a fraction of them was also given to an undergraduate student, who evaluated those independently. Finally, a graduate student in the lab spent some time counterchecking randomly-picked videos. No significant discrepancies were observed between these independent analyses. To quantify the speeds of these speckles, we generated kymographs that did not necessarily capture the whole journey of each speckle from the start to dissociation. The numbers obtained between different such kymographs were consistent with each other. We now explain this in the Materials and methods section in more detail.

3) Mutating G374 in Kip2 is a nice second independent test for whether Kip2 starts its runs from the microtubule minus-ends (subsection “Kip2 runs start from SPBs”, first paragraph). However, if Kip2 is indeed different from Kip3 with regard to where it initiates runs, then mutating the corresponding residue in Kip3 (G343 based on the alignment in Figure 3—figure supplement 1) would be a good control to perform. The mutated Kip3-G343 would be predicted to decorate along the microtubule length and not accumulate at the b-SPB like Kip2. This would strengthen the argument that loading at SPBs and initiating runs at the minus end is a Kip2-specific phenomenon.

We thank the reviewers for suggesting to use an ATPase deficient mutant of Kip3 as a control. We generated Kip3-G343A-3xsfGFP according to the alignment in Figure 3—figure supplement 1. As shown in Author response image 1, Kip3-G343A-3xsfGFP expressed from the endogenous locus did not bind to any specific structure in the cytoplasm; however, faint GFP fluorescence associated with nuclear microtubules (top panel). Next, we generated Kip3-E345A-3xsfGFP whose lack of microtubule stimulated ATPase activity has been experimentally validated (Arellano-Santoyo et al., 2017). Kip3-E345A-3xsfGFP did not associate with any specific structure at all (middle panel). We speculate that perturbing the switch-2 motif of Kip3 greatly reduces its affinity to microtubules. This was evident when we expressed Kip2-mCherry as the cytoplasmic microtubule marker in Kip3-E345A-3xsfGFP/Kip3 diploid cells. No visible Kip3-E345A-3xsfGFP/Kip3-E345A-3xsfGFP or Kip3-E345A-3xsfGFP/Kip3 dimers associated with cytoplasmic microtubules. Furthermore, very little ATPase deficient Kip3 molecules associated with nuclear microtubules (bottom panel).

Since the ATPase deficient mutants of Kip3 cannot serve as a control, we decided to not include these results in the revised manuscript.

4) In Figure 5B, the authors show that bfa1 and bub2 deletions lowered the levels of Kip2-G374A on the b-SPB by nearly 50%, while the signal on the m-SPB was unchanged. Similarly, in Figure 5F, the levels of Kip2 on b-microtubules lowered to nearly m-microtubules levels, while the signals on the m-microtubules remained unchanged compared to wild type. These decreases might indicate that there is actually less total Kip2 in bfa1 and bub2 null cells. The authors should determine the expression levels of Kip2 by western blot to exclude the possibility that the decreases are not due to a reduction in Kip2's expression or protein stability.

We performed Western blot analyses of Kip2-3xsfGFP in control cells as well as in *bfa1Δ, bub2Δ, bfa1Δ bub2Δ* cells. No change in Kip2-3xsfGFP protein abundance was observed. This result is now presented in Figure 2—figure supplement 3E.

This result is also consistent with the median protein concentration estimates given by the model, which do not show any substantial difference between wt and *bfa1Δ bub2Δ* cells (Figure 2—figure supplement 3A).

5) In a related point, following the analysis of Kip2-S63A-G374A in Figure 6C, D, E, the authors conclude that both SPBs recruited more of the hypo-phosphorylated, ATPase deficient protein and that the distribution was still asymmetric in cells with correctly oriented SPBs. However, based on Figure 6D, it seems that the enhancement in both SPBs indicates that the total amount of Kip2 in the cell might have increased significantly because of the S63A mutation. The authors should compare the levels of Kip2-S63A-G374A to Kip2-G374A to exclude the possibility that dephosphorylation of Kip2's N-terminus enhances the stability of the protein, thereby giving the observed effects.

We performed Western blot analyses of Kip2-G374A-3xsfGFP and Kip2-S63A-G374A-3xsfGFP. No change of Kip2 protein abundance was observed. This result is now presented in Figure 2—figure supplement 3E.

In addition, the median protein concentration estimates given by the model do not show any substantial difference between wt and Kip2-S63A cells (Figure 2—figure supplement 3A).

6) The authors' main conclusions concern Kip2's distribution profile in metaphase. However, they selected spindles for analysis and decided which spindles were in metaphase based on the shape of cells and the size of spindles (subsection “Image and data analysis”, first paragraph). The standard in the field for examining cells in metaphase is to perform cdc20 depletion (such as Khmelinskii et al., 2009). Given that metaphase is not strictly defined by the length of the spindle, it would be more appropriate to refer to preanaphase instead of metaphase.

This is a good point. We have now changed our wording accordingly.

7) Kip2 hypo-phosphorylation caused an increase in the levels of Kip2 on the plus ends of both b- and m-microtubules compared to wild type, based on Figure 6J versus Figure 4E, but the levels of dynein to the b-microtubules, a cargo of Kip2, did not appear to increase based on the images shown in Figure 7C (comparing the dynein dots in KIP2 vs. KIP2-S63A images). On the other hand, in Figure 7D, the authors report that dynein distribution at the tip of b- and m-microtubules was randomized when Kip2 phosphorylation is prevented (i.e. in KIP2-S63A mutant cells). Please clarify how randomization occurred when there is no apparent change in dynein intensity?

We suggest that the total amount of dynein being limiting explains these observations. As shown in Figure 7C, the levels of Dyn1-mNeonGreen do not appear to increase on the b-microtubules in Kip2-S63A cells. Based on our data, the randomization in Kip2-S63A cells is achieved through redistribution of dynein between b- and m-microtubules. Pairwise analysis of the Dyn1-mNeonGreen intensity between b- and m- microtubules in wt and Kip2-S63A cells is shown in Figure 7—figure supplement 2. For cells nucleating both b- and m-microtubules, increasing the levels of Kip2 on both microtubules (i.e., in Kip2-S63A cells) leads to a strong redistribution of dynein from b- to m-microtubules. In the graph, mean ± s.d. of the normalized intensities are shown in red. We have now added Figure 7—figure supplement 2 and added a sentence in the main text to state our finding.

8) Related to the modeling.- The presented model addresses binding, unbinding and stepping rates of the kinesin. However, it does not consider dynamic microtubules (e.g. compare left vs. right kymographs in Figure 3A, where the plus end clearly grows in the experimental condition, but not in the model). Addition of plus end dynamics and showing that the regulation of Kip2 loading at the minus ends affects it would make the model predictions much stronger. For instance, how would the model predict observations presented in Figure 4?

We agree with the reviewers that a more comprehensive model incorporating microtubule dynamics would allow addressing very directly how the flux of Kip2 coming from the minus end affects microtubule dynamics at the plus end. However, it would not address in a more substantial manner the first and main point of the manuscript, which is the fact that Kip2 starts its runs nearly exclusively from the minus end of the microtubule. For this purpose, our original model has two main advantages:

A) Constructing the model using a fixed (but arbitrary) length allowed us to make analytical predictions about the parameter values that would allow for a flat Kip2 distribution profile, without resorting to numerical simulation. These predictions were instrumental in guiding the design of our experiments.

B) Kip2 moves towards microtubule plus ends substantially faster than microtubules grow at the plus ends (approximately by a factor of 4 in our data, Figure 2—figure supplement 1B, D, consistent with approximately a factor of 4 in the in vitro data of Hibbel et al., 2015) – thus we did not expect to see a major difference if microtubule growth is included as a parameter in the model. Our original model is therefore a good compromise, which is supported by the extended model analysis below.

However, the comment of the reviewers made us realize that growing microtubules could affect the accumulation of Kip2 at the microtubule plus end. To investigate this possibility, we prescribed microtubule growth in a novel version of the model at the in vivomeasured growth rates. While microtubule growth does not change the lattice and peak intensity of the predicted mean Kip2 distribution profile substantially, it does result in a wider Kip2 peak at the microtubule end (see Figure 2—figure supplement 2: panel A corresponds to the analysis as performed, panel B is a control with the growing model and extremely slow growth, similar to no growth, and panel c shows the simulation results with growth at in vivo microtubule growth speeds). This would lead to a better fit in Figure 2B, since the left “shoulder” of the Kip2 signal at the plus end is specifically one area where the original model shows mismatches with the experimental data. From this data, we conclude that a model that incorporates the measured microtubule dynamics will not change the model predictions qualitatively in such a way that it affects the findings on the remote-control mechanism. We have now added Figure 2—figure supplement 2, as well as a short statement on the findings to the main text. In addition, we provide the code to run such a model in the updated version of our GitLab repository for the paper, so that other researchers interested in these tip effects can re-use our analysis.

For what concerns modeling of the effect of Kip2 flux on microtubule dynamics, this is a completely different undertaking that is out of reach for now. Such an extended model would require dose-response-type data on instantaneous microtubule growth rates given the Kip2 intensity at the microtubule plus tip, which would require dynamic data reporting concomitantly on microtubule dynamics and Kip2 levels. For a number of reasons, we do not have these data available at this point. The intractability of in vivo kymographs due to the movement of microtubules in vivo, as discussed in our answer to point 1, is one of the issues that we have not routinely solved yet. While we agree with the reviewers that such a model would be very desirable, the required effort for its development is beyond the scope of the current study (and would not contribute to supporting this study’s key findings in our view).

- The total intensity at the tip corresponds to a defined accumulation of motors at the end. How many are there? (The number of occupied sites should be set by the ratio of minus-end loading and plus-end off rate; and can thus also be directly predicted from the model parameters.)

This depends on the definition of “tip”. If we define the tip as the terminal site of each protofilament, it is bound on average 76% of the time at the median parameter values we report in Figure 2C, D, E, corresponding to 10 molecules of Kip2 (assuming 13 protofilaments). Another approach would be to define the tip as the high-intensity area at the plus end, corresponding to approximately the last 500 nm of the microtubule (Figure 1B). According to this definition, at the median parameter values we report in Figure 2C, D, E, approximately 390 molecules of Kip2 are present at the tip. We have added the code to compute these values to the GitLab repository that accompanies the manuscript.

9) Points to discuss:- Is it possible to exclude direct end-targeting of Kip2 (e.g. through Bim1 binding or another mechanism)? In principle, such a model would also produce a non-microtubule-length dependent Kip2 profile, here the r_on_ would be the on rate of direct binding to the tip.

Microtubule associated proteins like Bim1 and Bik1 are enriched at the plus-ends of cytoplasmic microtubules. Both Bim1 and Bik1 have affinity to Kip2. Yet, Kip2 is not directly targeted to microtubule plus ends. As shown in Figure 3C, the ATPase-deficient mutant protein Kip2-G374A-3xsfGFP is not present along microtubule shafts or at microtubule plus ends. This is explained in detail in the second paragraph of the subsection “Kip2 runs start from SPBs”. This is also the reason why no plus-tip-specific binding rate (*r_on_+*) was included in the model. We have added a sentence to make this point in corresponding part of the Results section.

- If enhanced loading to the SPB is a specific mechanism for the b-SPB and not the m-SPB, the prediction would be that the distribution of Kip2 intensities is length-dependent on m-SPB-emanating microtubules. Is that true?

Our model would predict that the shape of the Kip2 profile (plateau along the microtubule shafts and microtubule length-independent Kip2 intensity at the plus end) would be similar for microtubules emanating from the m-SPB and the b-SPB, as long a Kip2 recruitment is confined to the SPB. The data in Figure 5—figure supplement 1B are consistent with this prediction (cf. also Figure 5—figure supplement 1B for the Kip2-S63A mutant with lattice loading and length-dependence for both SPB microtubules). For the differentiation between m-SPB and b-SPB, the flux coming from the m-SPBs is not turned off, it is simply much lower than that coming from b-SPBs, leading to similar profile shapes but lower overall intensities of Kip2 for m-SPBs (see Figure 5—figure supplement 1B).